# Dual stimuli-responsive rotaxane-branched dendrimers with reversible dimension modulation

Xu-Qing Wang[1], Wei Wang[1], Wei-Jian Li[1], Li-Jun Chen[1], Rui Yao[1], Guang-Qiang Yin[1,2], Yu-Xuan Wang[1], Ying Zhang[3], Junlin Huang[4], Hongwei Tan[3], Yihua Yu[4], Xiaopeng Li[2], Lin Xu[1] & Hai-Bo Yang[1]

With the aim of mimicking biological machines, in which the delicate arrangement of nanomechanical units lead to the output of specific functions upon the external stimulus, the construction of dual stimuli-responsive rotaxane-branched dendrimers was realized in this study. Starting from a switchable organometallic [2]rotaxane precursor, the employment of a controllable divergent approach allowed for the successful synthesis of a family of rotaxane-branched dendrimers up to the third generation with 21 switchable rotaxane moieties located on each branch. More importantly, upon the addition and removal of dimethylsulfoxide (DMSO) molecule or acetate anion as the external stimulus, the amplified responsiveness of the switchable rotaxane units endowed the resultant rotaxane-branched dendrimers the solvent- or anion-controlled molecular motions, thus leading to the dimension modulation. Therefore, we successfully constructed a family of rotaxane-branched dendrimers with dual stimuli-responsiveness that will be a privileged platform for the construction of dynamic supramolecular materials.

[1] Shanghai Key Laboratory of Green Chemistry and Chemical Processes, School of Chemistry and Molecular Engineering, Chang-Kung Chuang Institute, East China Normal University, Shanghai 200062, P.R. China. [2] Department of Chemistry, University of South Florida, Tampa, FL 33620, USA. [3] Department of Chemistry, Beijing Normal University, Beijing 100050, P.R. China. [4] Shanghai Key Laboratory of Magnetic Resonance, Department of Physics, East China Normal University, Shanghai 200062, P.R. China. These authors contributed equally: Xu-Qing Wang, Wei Wang. Correspondence and requests for materials should be addressed to L.X. (email: lxu@chem.ecnu.edu.cn) or to H.-B.Y. (email: hbyang@chem.ecnu.edu.cn)

**M**echanically interlocked molecules (MIMs), such as rotaxanes, catenanes, knots, etc., have aroused extensive interests during past few decades because of not only their esthetic beauty but also their extensive applications as artificial molecular machines[1–6]. For instance, the pioneering work on rotaxane-based molecular shuttles by 2016 Nobel laureate, J. Fraser Stoddart, has initiated a new era of design and synthesis of molecular machines[7,8]. The inspiration of the construction of MIM-based artificial molecular machines comes from the living systems, in which well self-organization of functional nanomechanical moieties enables the intriguing amplification of collective molecular motions to perform vital biological functions[9–11]. As a representative example, the macroscopic motion of muscles is realized by the coordinative movements of sarcomeres as repeating units[12]. By mimicking the delicate arrangement of nanomechanical units in biological machines, the introduction of artificial MIMs into a specific supramolecular scaffold will inject new vitality to the construction of new dynamic supramolecular materials[13,14].

Rotaxanes, as a fundamental type of MIMs, have proven to be crucial candidates for the construction of artificial molecular machinery and electronic devices due to their shuttling and switching features[15–18]. Upon being exposed to the external stimulus such as pH, redox, temperature, light etc., the macrocyclic component in rotaxane could undergo directional motions around the axle component[19]. By combining such unique motion properties of rotaxane with the monodispersed and highly symmetrical nature of dendrimers[20–22], investigations on rotaxane dendrimers have offered great possibilities towards the construction of novel smart materials, which have attracted considerable attentions recently[23,24]. Up to date, a great number of sophisticated rotaxane dendrimers have been successfully prepared by Vögtle and co-workers[25], Stoddart and co-workers[26–28], Gibson et al.[29], Kim and co-workers.[30,31], Wang and Kaifer[32], Leung and co-workers[33,34], etc., which have displayed wide applications in the field of molecular nanoreactors, gene delivery, and light-harvesting system, etc.[35–37]. It should be noted that, although a great deal of achievement has been obtained in this field, the rotaxane-branched dendrimers, in which rotaxane moieties are located on each branch, have been rarely explored because of the synthetic chanllenge[31,33,34,38]. In particular, the construction of multiple stimuli-responsive rotaxane-branched dendrimers has not been yet realized.

Previously, we have successfully synthesized the high-generation (up to fourth generation) organometallic rotaxane-branched dendrimers via a controllable divergent strategy[38]. However, in that example, the lack of controllable switching property of the rotaxane units hampered the investigation on their stimuli-responsiveness and further applications.

Herein, as a feasible and practical solution towards the construction of dynamic rotaxane-branched dendrimers as smart supramolecular materials, we describe herein the preparation and characterization as well as property investigation of a family of dual stimuli-responsive rotaxane-branched dendrimers up to third generation with 21 switchable [2]rotaxane units located on each branch. The introduction of switchable rotaxane units into the dendritic scaffold imparted the switchable feature to the resultant rotaxane-branched dendrimers when the external stimulus was added. For example, with the addition of dimethylsulfoxide (DMSO) molecule or acetate anion as the competitive hydrogen bonding acceptor, the directional and switchable mechanical motion of rotaxane on each branch was realized, leading to the dynamics and dimension modulation of the integrated rotaxane-branched dendrimers (Fig. 1). Therefore, the controllable mechanical motions of rotaxane moieties could influence the dynamics and dimensions of rotaxane-branched dendrimers, which might be applied in the reversible uptake and release applications, or even switchable organocatalysis in the future.

## Results

**Synthesis of the switchable [2]rotaxane 2 as key precursor.** In this study, the host–guest complex of pillar[5]arene and neutral alkyl chain was employed as the rotaxane moiety[39–41]. Meanwhile, the formation of platinum–acetylide bond was selected as the key growth steps[42,43]. In order to realize the controllable, directional motion of macrocycle to afford the switchable rotaxane, the urea moiety was inserted into the backbone of axle to serve as stimuli-responsive site in the pillararene/alkyl chain rotaxane system. Due to the stronger hydrogen bonding interactions between ethoxy group of pillararene macrocycle and the urea moiety compared with the CH···π interactions between pillararene macrocycle and neutral alkyl chain, the urea moiety and

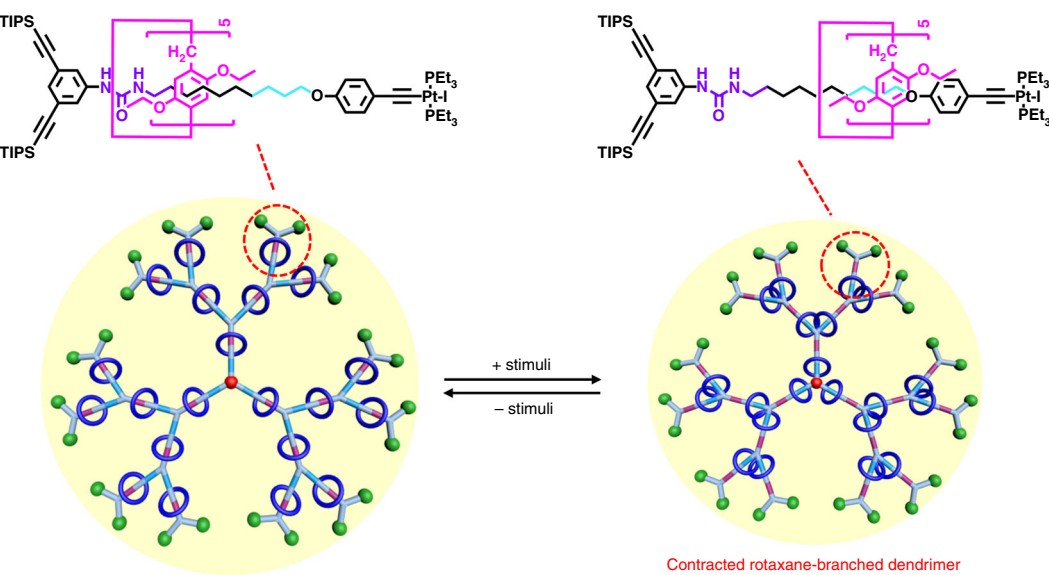

**Fig. 1** Cartoon representation of the dimensional modulation of rotaxane-branched dendrimer upon the addition or removal of external stimuli

the adjacent methylene units would be preferentially positioned within the cavity of the pillar[5]arene. Upon the addition of hydrogen bonding acceptor as stimulus to the pillararene/alkyl chain rotaxane system containing the urea group, the controllable motion of pillar[5]arene to methylene units will be feasible, thus resulting in the construction of a switchable rotaxane (Fig. 2). According to such design strategy, a semi-blocked rod-like component **1** possessing both urea moiety and neutral alkyl chain, respectively, was prepared by a multistep synthetic route as indicated in SI Appendix (Supplementary Fig. 1). In the presence of CuI as a catalyst, the mixture of semi-blocked rod-like component **1**, DEP5 (1,4-diethoxypillar[5]arene) macrocycle, and Pt(PEt$_3$)$_2$I$_2$ in a ratio of 1:6:4 in CHCl$_3$/$i$-Pr$_2$NH (v/v, 2:1) led to the successful synthesis of [2]rotaxane **2** as the key precursor for the following dendrimer growth in a good yield (70%) on gram scale.

The analysis of multinuclear ($^1$H, $^{13}$C, and $^{31}$P) NMR revealed the formation of organometallic [2]rotaxane **2**. As shown in the $^1$H NMR spectrum (Supplementary Fig. 21), due to the shielding effect, the peaks of urea (H$_3$ and H$_4$) and methylene moiety (H$_{8-17}$) on the axle component displayed remarkable upfield shifts, and two broad peaks below zero were found. All these observations indicated the successful formation of the pillararene/alkyl chain rotaxane system. With the assistance of two-dimensional (2-D) spectroscopic techniques ($^1$H-$^1$H COSY and ROESY), the formation of organometallic [2]rotaxane **2** was further confirmed. By virtue of 2-D COSY analysis (Supplementary Fig. 23), the signals of the alkyl chain were clearly identified. Meanwhile, in the 2-D ROESY spectrum (Supplementary Fig. 24),

the correlations between proton H$_4$ of urea moiety on the axle with the aromatic protons (H$_a$) and the bridged methylene protons (H$_c$) of pillar[5]arene were observed. In addition, the methylene protons (H$_{8-17}$) of axle component also displayed correlations with protons H$_a$ and H$_c$ of pillar[5]arene, thus indicating the existence of the targeted rotaxane. More importantly, it was found that the correlations between protons H$_{9-13}$ and protons of pillar[5]arene (H$_a$ and H$_c$) were stronger than other protons in the axle, which demonstrated that these protons were encapsulated within the cavity of pillar[5]arene. Moreover, in $^{31}$P NMR spectrum (Supplementary Fig. 22), compared with that of Pt(PEt$_3$)$_2$I$_2$, the peak attributed to the phosphine ligand in [2]rotaxane **2** shifted from 1.09 to 9.55 ppm, which was consistent with the formation of platinum–acetylide bond. The study of MALDI-TOF-MS provided further strong evidence for the existence of organometallic [2]rotaxane **2**. In the mass spectrum of **2**, a peak at $m/z = 2201.0498$ was observed, which was attributed to [M + H]$^+$ ion. This peak was isotopically resolved and its isotopic resolution agreed well with the theoretical distribution (Supplementary Fig. 20). Notably, unlike some classic charged rotaxane systems based on either the charged macrocycles or axles, the organometallic [2]rotaxane **2** is neutral, which is able to simplify the subsequent reaction and purification processes of the synthesis of rotaxane-branched dendrimers.

## Synthesis and characterization of rotaxane-branched dendrimers.
With the key precursor [2]rotaxane **2** in hand, the

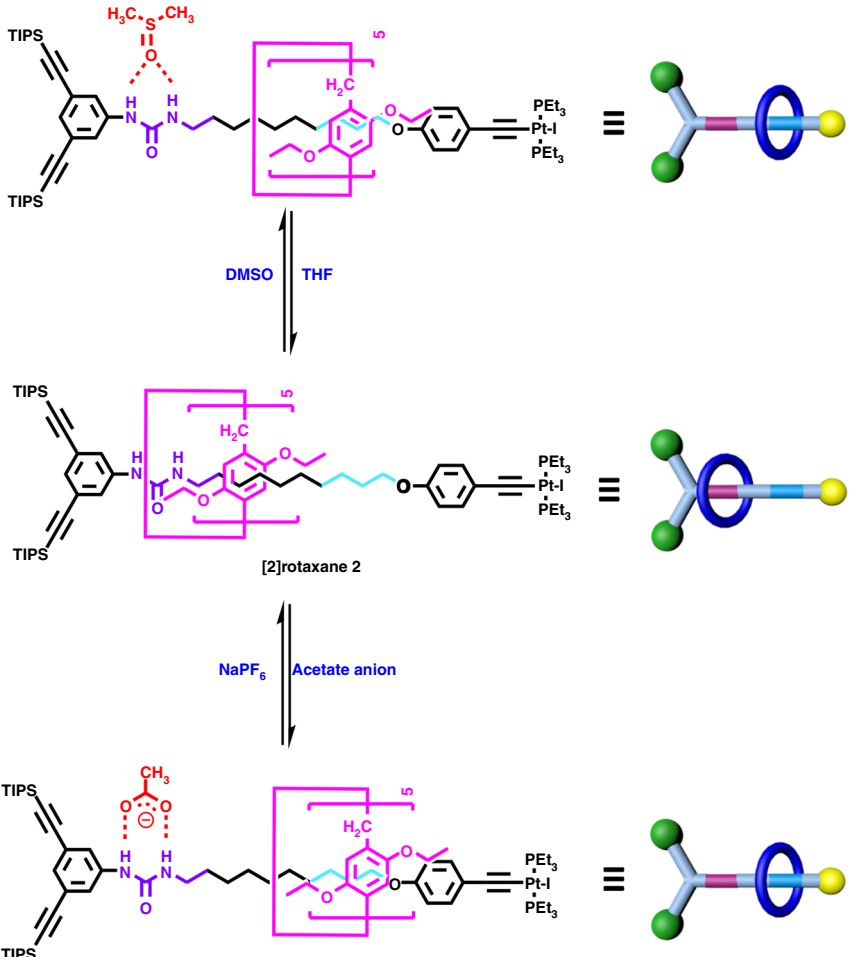

**Fig. 2** Cartoon representation of the solvent- and anion-induced switching motions of DEP5 ring in [2]rotaxane **2**

synthesis of rotaxane-branched dendrimers, in which pillararene/alkyl chain rotaxane system is located on each branch, was then performed. As shown in Fig. 3, by employing CuI-catalyzed coupling reaction of [2]rotaxane precursor **2** with 1,3,5-triethynylbenzene, the first-generation rotaxane-branched dendrimer **G₁** was successfully prepared in a yield of 74%, which contained three switchable rotaxanes on the branches. The sequential deprotection of **G₁** with tetrabutylammonium fluoride (TBAF) gave rise to the corresponding rotaxane dendrimer **G₁-YNE** with six alkyne groups at the periphery in a yield of 92%. By repeating the coupling reaction that generated platinum–acetylide bonds, the second-generation rotaxane-branched dendrimer **G₂** with nine switchable rotaxanes on the branches was synthesized in 62% yield. Similarly, the third-generation rotaxane-branched dendrimer **G₃** was prepared via the sequential deprotection-coupling process. It should be mentioned that the resultant third-generation rotaxane dendrimer **G₃** was a highly branched [22] rotaxane system with 21 rotaxane moieties located in the dendrimer skeleton of monodispersed distribution. The purification of these rotaxane-branched dendrimers **G₁−G₃** was performed via column chromatography and preparative gel permeation chromatography (GPC) (Supplementary Fig. 28).

Multinuclear ($^1$H, $^{31}$P, and $^{13}$C) NMR measurements were firstly performed to characterize the resultant rotaxane-branched dendrimers. In view of $^1$H NMR analysis (Fig. 4), the peaks of protons ascribed to the rotaxane units remained, which disclosed

that the rotaxane units were not destroyed during the growth process. Notably, in the higher-generation rotaxane-branched dendrimers **G₂** and **G₃**, the peaks became broad and more than one set of peaks attributed to the rotaxane units were observed, which indicated that the rotaxane moieties on different branches were slightly nonequivalent. In addition, $^{31}$P NMR spectra of all rotaxane-branched dendrimers **G₁−G₃** displayed a signal peak, which was consistent with the high symmetry feature of the dendritic skeleton. Similar with $^1$H NMR spectra, along with the generation increase of rotaxane-branched dendrimers, slight broad effect in the $^{31}$P NMR spectra was observed (Fig. 4). Moreover, compared with the building block [2]rotaxane **2**, the phosphine signals attributed to the PEt₃ ligands around platinum centers displayed the similar downfield shift of ~2.9 ppm (Supplementary Fig. 50), which provided the direct supports for the formation of platinum–acetylide bonds during the dendrimer growth process.

With the assistance of the mass analysis, the formation of rotaxane-branched dendrimers was further confirmed. In the ESI-MS spectrum of **G₁** (Supplementary Figs. 32 and 33), peaks of $m/z = 2124.7927$ and $m/z = 1593.9254$ were found, which agreed well with the theoretical value of $[G_1 + 3\,H]^{3+}$ ion ($m/z = 2124.8766$) and $[G_1 + 4\,H]^{4+}$ ion ($m/z = 1593.9075$). MALDI-TOF-MS experiment was further performed to analyze the structure of **G₂**. The peak of $m/z = 17944.7$ for **G₂** was observed, which was almost in consistent with the theoretical average

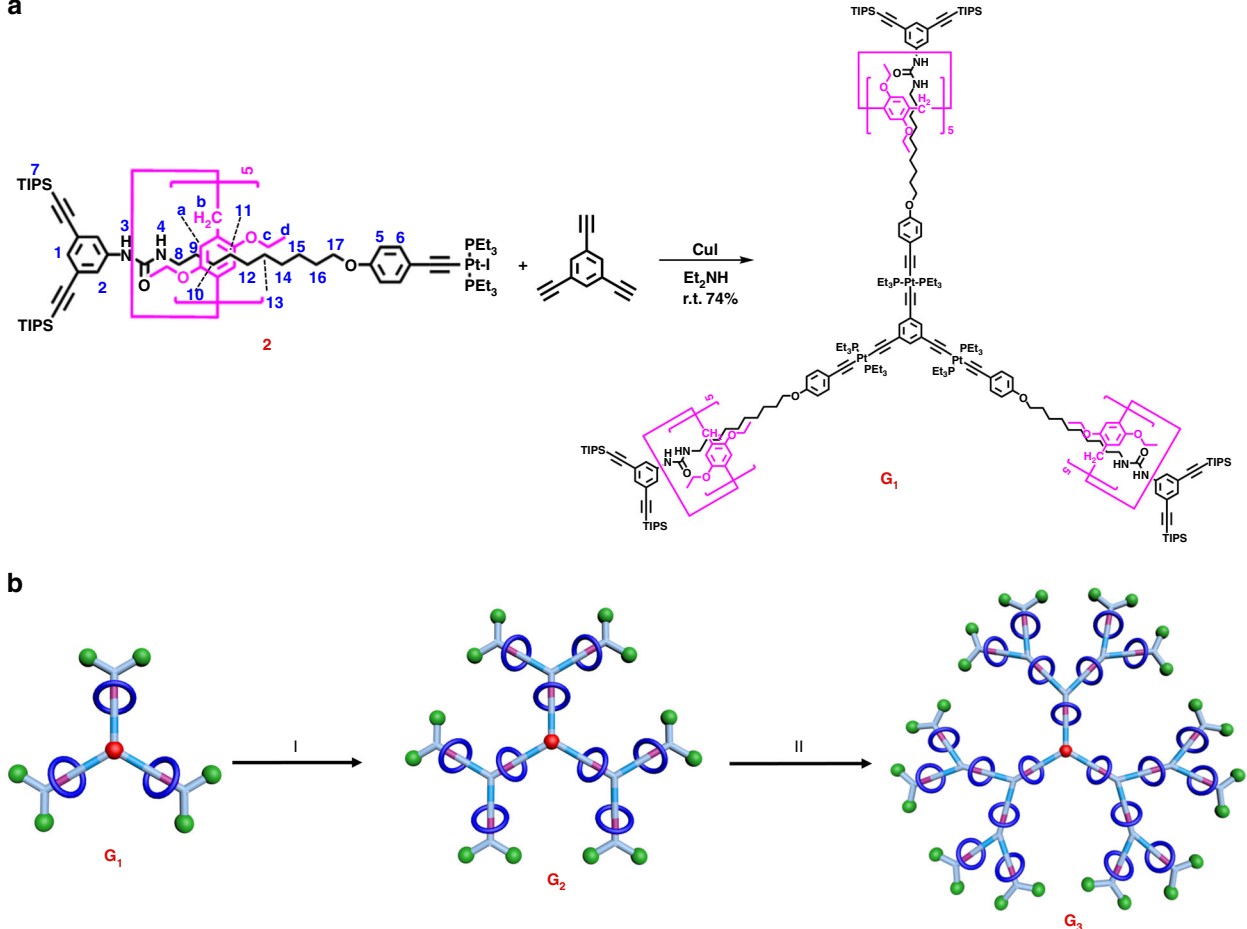

**Fig. 3 a** Synthesis of rotaxane-branched dendrimer **G₁** by a CuI-catalyzed coupling reaction of [2]rotaxane building block **2** and 1,3,5-triethynylbenzene; **b** schematic representation of a controllable divergent approach for the synthesis of rotaxane-branched dendrimers **G₂** and **G₃**. Reaction conditions: (I) (a) TBAF, THF, r.t., 4 h, 92%; (b) **2**, CuI, Et₂NH, r.t., 8 h, 62%; (II) (a) TBAF, THF, r.t., 4 h, 60%; (b) **2**, CuI, Et₂NH, r.t., 8 h, 61%

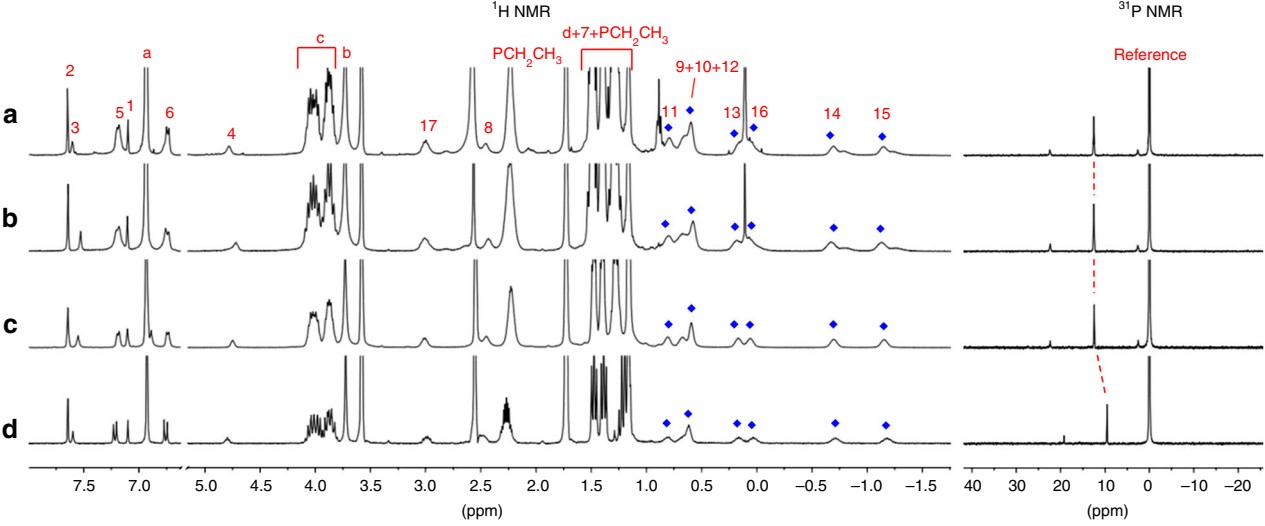

**Fig. 4** Partial $^1$H NMR (THF-$d_8$, 298 K, 400 MHz) and $^{31}$P NMR (THF-$d_8$, 298 K, 122 MHz) spectra of **a** rotaxane-branched dendrimer **G$_3$**; **b** rotaxane-branched dendrimer **G$_2$**; **c** rotaxane-branched dendrimer **G$_1$**; **d** [2]rotaxane **2**

molecular weight $M_r = 17876.5$ Da (Supplementary Fig. 42). Due to the large molecular mass and low ionization efficiency of **G$_3$**, neither MALDI-TOF-MS nor ESI-MS offered the satisfied mass data. GPC experiments were then carried out to confirm the formation as well as the monodispersity of rotaxane-branched dendrimers. In the GPC spectra (Supplementary Figs. 51–53), all rotaxane-branched dendrimers exhibited a single peak and narrow distributions for the number-averaged molecular weight (Mn) and the polydispersity index (PDI) (for **G$_1$**, PDI = 1.03; for **G$_2$**, PDI = 1.04; for **G$_3$**, PDI = 1.15), indicating the existence of monodisperse rotaxane-branched dendrimers **G$_1$**–**G$_3$**.

Moreover, 2-D diffusion-ordered spectroscopy (DOSY)[44–47] was also exploited to evaluate the monodispersity and size change of the resultant rotaxane-branched dendrimers **G$_1$**−**G$_3$**. All DOSY spectra of **G$_1$**–**G$_3$** presented one set of signals, indicating the existence of the sole species of rotaxane-branched dendrimers. Furthermore, compared with the key [2]rotaxane building block **2**, the significant decrease of the diffusion coefficient (D) from $(15.15 \pm 0.05) \times 10^{-10}$ m$^2$ s$^{-1}$ (**2**) to $(7.37 \pm 0.04) \times 10^{-10}$ m$^2$ s$^{-1}$ (**G$_1$**), $(5.69 \pm 0.06) \times 10^{-10}$ m$^2$ s$^{-1}$ (**G$_2$**), and even $(2.95 \pm 0.07) \times 10^{-10}$ m$^2$ s$^{-1}$ (**G$_3$**), respectively, were clearly observed, which provided the additional support for the progressive size increase of the obtained rotaxane-branched dendrimers (Supplementary Figs. 54–57).

With the targeted rotaxane-branched dendrimers in hand, the investigations on their morphology and photophysical properties were further carried out. Atomic force microscopy (AFM) and transmission electron microscopy (TEM) were employed to study the morphology of the resultant dendrimers **G$_1$**−**G$_3$**. According to the AFM images (Supplementary Figs. 58–60), it was found that, with the increase of generation of rotaxane-branched dendrimers, the average height gradually increased from $1.76 \pm 0.26$ nm (**G$_1$**) to $2.70 \pm 0.29$ nm (**G$_2$**), and $3.21 \pm 0.34$ nm (**G$_3$**), respectively. From the TEM analysis, the size of dendrimers **G$_1$**−**G$_3$** were determined to be $1.86 \pm 0.24$ nm for **G$_1$**, $2.60 \pm 0.27$ nm for **G$_2$**, and $3.55 \pm 0.32$ nm for **G$_3$**, respectively (Supplementary Figs. 61–63). In the UV-vis spectra (Supplementary Table 1 and Supplementary Fig. 64), two major absorption bands at around 291 and 343 nm were found, which were described as an admixture of intra-ligand (IL) [π–π* (CR≡CR)] and metal-to-ligand charge transfer (MLCT) [d(Pt)π–π* (CR≡CR)] transition with the predominant IL character according to the previous spectroscopic investigation on platinum–acetylide complexes[48–50].

**Solvent- or anion-induced switching of [2]rotaxane 2.** With the successful construction of the targeted rotaxane-branched dendrimers **G$_1$**−**G$_3$**, the investigations on their responsiveness towards the external stimuli were on the agenda. As the key building block for preparation of the integrated rotaxane-branched dendrimers, the stimuli-responsive behavior of [2] rotaxane **2** was firstly evaluated. Due to the existence of urea moiety as a hydrogen bonding donor, a DMSO molecule or an acetate anion was employed as a hydrogen bonding acceptor to compete with a DEP5 macrocycle to complex with the urea group. Therefore the solvent- or anion-induced translational motion of DEP5 macrocycle in [2]rotaxane **2** should be feasible (Fig. 2). For instance, by sequentially adding DMSO-$d_6$ into the solution of [2]rotaxane **2** ($c = 0.4$ mM) in tetrahydrofuran-$d_8$ (THF-$d_8$), the DEP5 ring gradually moved from the urea moiety towards the neutral alkyl chain, which was indicated by the $^1$H NMR titration experiments as shown in Supplementary Table 2 and Supplementary Fig. 80. It was found that the signals of the protons H$_3$ and H$_4$ on the urea moiety as well as methylene protons (H$_{8–12}$) nearby the urea moiety were all shifted downfield. On the contrary, the signals of protons (H$_{13–17}$) ascribed to the neutral alkyl chain displayed the obvious upfield shifts. For example, with the continuous increase of DMSO-$d_6$, the signal of proton H$_3$ shifted downfield from 7.50 ppm (THF-$d_8$, 400 μL) to 8.22 ppm (THF-$d_8$/DMSO-$d_6$, 400/10 μL). On the other hand, the signal of the proton H$_{15}$ moved upfield from −1.07 ppm (THF-$d_8$, 400 μL) to −1.78 ppm (THF-$d_8$/DMSO-$d_6$, 400/10 μL). Moreover, the further addition of THF-$d_8$ (100 μL) into the system induced the translational motion of DEP5 ring in the opposite direction. During such directional motion process, the signal of protons H$_{3–4}$ and H$_{8–12}$ shifted upfield, whereas the signal of proton H$_{13–17}$ shifted downfield, thus indicating the reversibility of solvent-induced translational motion behavior of DEP5 macrocycle in [2]rotaxane **2**.

Similarly, the sequential addition of tetrabutylammonium acetate (TBAA) into the solution of [2]rotaxane **2** in THF-$d_8$ induced the significant changes in the $^1$H NMR spectra as indicated in Supplementary Table 3 and Supplementary Fig. 81. The signals of the protons H$_3$ and H$_4$ on the urea moiety were remarkably downfield shifted. At the same time, the signals of aromatic protons (H$_1$ and H$_2$) and methylene protons (H$_{8–12}$) close to the urea moiety also shifted downfield, while the signal of protons (H$_{13–17}$) ascribed to the neutral alkyl chain exhibited the

obvious upfield shift. For example, upon adding 5.0 equiv. of TBAA, the signal of proton $H_3$ shifted downfield from 7.48 to 11.71 ppm. On the opposite, the peaks of $H_{15}$ moved upfield from $-1.05$ ppm to $-2.11$ ppm, suggesting that the DEP5 macrocycle moved away from the urea moiety towards the neutral alkyl chain. In order to completely remove acetate anion as NaOAc precipitate, 7.0 equiv. of $NaPF_6$ was subsequently added into the mixture of [2]rotaxane 2 and TBAA. The resultant $^1H$ NMR spectrum was almost the same as the original spectrum of the [2]rotaxane 2, indicating that the DEP5 macrocycle moved back to the urea moiety. With the aim to obtain the further insight into the anion-induced switching of [2]rotaxane 2, the acetate binding affinity of [2]rotaxane 2 was determined by $^1H$ NMR titrations with acetate anion (TBAA) in THF-$d_8$. The data were fitted to a 1:1 binding model (2: acetate anion) as confirmed by Job plot analysis, and the anion binding constant was calculated to be log $K = 3.57 \pm 0.2$ (Supplementary Fig. 85).

It should be noted that, in order to provide the additional support to such stimuli-induced switching behavior of [2]rotaxane 2, a series of control experiments were carried out. Two model complexes either without the urea moiety (2-a) or without the pillar[5]arene macrocycle (2-b) were synthesized as shown in Supplementary Figs. 65 and 66. In the case of model complex 2-a without urea moiety, upon the addition of DMSO-$d_6$ (10 μL) or TBAA (5.0 eq.) into the solution of 2-a in THF-$d_8$ (9.0 mM, 400 μL), the resultant spectra showed no obvious change compared with the original spectrum of 2-a as indicated in $^1H$ NMR spectra (Supplementary Fig. 86). While for the [2]rotaxane 2, upon the addition of either DMSO-$d_6$ (10 μL) or TBAA (5.0 eq.) as the stimulus at the same concentration (0.4 mM), the obvious downfield shifts of the protons on urea moiety ($H_3$ and $H_4$) were observed (Supplementary Fig. 87). The combination of these findings suggested that the urea moiety did act as a binding site interacting with the DMSO molecule or anion species. More importantly, by comparing the $^1H$ NMR spectra of model complex 2-b with the [2]rotaxane 2 before and after the addition of DMSO molecule or anion species, the translational motion of DEP5 macrocycle along the axle was confirmed. When comparing the $^1H$ NMR spectrum of [2]rotaxane 2 with the one of model complex 2-b (Supplementary Figs. 88b, c and 89b, c), the obvious upfield shifts of urea protons ($H_3$ and $H_4$) were observed in the $^1H$ NMR spectrum of [2]rotaxane 2, thus suggesting the encapsulation of urea moiety and the adjacent methylene units within the cavity of DEP5. While upon the addition of DMSO molecule as stimulus, as indicated in the $^1H$ NMR spectra of the model complex 2-b and the corresponding [2]rotaxane 2 (Supplementary Figs. 88a and 88d, respectively), the protons on urea moiety ($H_3$ and $H_4$) downfield shifted to the almost same position. In couple with the obvious upfield shifts of the methylene units ($H_{13}$, $H_{14}$, $H_{15}$, and $H_{16}$) in the axle of [2]rotaxane 2, the stimuli-induced movement of DEP5 macrocycle from the urea moiety to the neutral alkyl chain on the other side was demonstrated. In the case of acetate anion as stimulus, the same results were observed as shown in Supplementary Fig. 89. These results strongly supported the existence of translational motion of DEP5 macrocycle along the axle stimulated by DMSO molecule or acetate anion.

In order to gain a better understanding of the solvent- and anion-induced switching motion of DEP5 ring in [2]rotaxane 2, theoretical calculation on these complexes was performed by using the PM6 method with Grimme's D3 correction for correlation with MOPAC2016 program package[51,52]. As shown in Supplementary Fig. 90, the optimized geometry structure indicated that, in the initial state, the methylenes protons $H_8$ and $H_9$ near the urea moiety were encapsulated within the aromatic cavity of DEP5 ring. Upon the addition of the DMSO molecule as

stimulus, due to the formation of hydrogen bonding complexes with the urea moiety, the DEP5 ring moved away and located around the methylenes with protons $H_{14}$ and $H_{15}$. Similarly, when the acetate anion was added as stimulus, the DEP5 ring underwent the similar translational motion to the methylenes with protons $H_{15}$ and $H_{16}$. Notably, in the case of the acetate anion as stimulus, the significant anion-induced folding of the larger stopper site in rotaxane was observed, which laid the foundation for the further dimension modulation of integrated rotaxane systems. According to the aforementioned results, the solvent- and anion-controlled translational motion of the DEP5 ring in [2]rotaxane 2 was confirmed, just allowing for construction of dynamic rotaxane-branched dendrimers by employing [2]rotaxane 2 as the precursor.

**Solvent- or anion-induced switching of rotaxane-branched dendrimers**. On the basis of the aforementioned switching motion of DEP5 in [2]rotaxane 2 stimulated by either DMSO molecule or acetate anion, the amplification effect of multiple switchable rotaxanes in the integrated rotaxane-branched dendrimers $G_1$−$G_3$ with the same stimulus was then investigated in detail. It was found that, upon the progressive addition of DMSO-$d_6$ into the THF-$d_8$ solution of rotaxane-branched dendrimers, the DEP5 rings in all dendrimers $G_1$−$G_3$ displayed the similar solvent-induced switching motions as indicated by $^1H$ NMR titration experiments (Supplementary Table 4 and Supplementary Fig. 92 for $G_1$; Supplementary Table 6 and Supplementary Fig. 108 for $G_2$; Fig. 5, Supplementary Table 8 and Supplementary Fig. 111 for $G_3$). Obvious chemical shifts of the related proton signals on the axle were observed, which displayed the same trend with [2]rotaxane building block 2, thus indicating the feasibility of solvent-induced switching behavior of the integrated rotaxane-branched dendrimers.

Moreover, $^1H$ NMR titration experiments via adding the increase amount of acetate anions suggested the anion-controlled switching behavior of the resultant rotaxane-branched dendrimers as well. According to the detailed titration study of switchable [2]rotaxane 2, for each urea moiety, 5.0 equiv. of TBAA was employed to induce the translational motion of DEP5 rings and 7.0 equiv. of $NaPF_6$ was required to return the dendrimers to the original state. In the case of first-generation rotaxane-branched dendrimer $G_1$, the $^1H$ NMR titration experiment indicated that the addition of 15.0 equiv. of TBAA resulted in the complete switching motion of DEP5 rings (Supplementary Table 5 and Supplementary Fig. 93). Notably, Job plot analysis indicated a 1:3 ($G_1$: acetate anion) binding stoichiometry (Supplementary Fig. 94a), and the binding constants for the acetate anion were calculated to be log $K_1 = 4.19 \pm 0.2$, log $K_2 = 3.43 \pm 0.2$, log $K_3 = 3.21 \pm 0.2$ (Supplementary Fig. 94b). Moreover, after adding 21.0 equiv. of $NaPF_6$ into the mixture of $G_1$ and TBAA, the corresponding proton signals returned to the initial position, suggesting that DEP5 rings went back to their original positions.

Considering the fact that different locations of DEP5 ring on the branches may influence their folding behaviors, which could further lead to the size modulation of rotaxane-branched dendrimers, thus the detailed investigation on the anion-induced dimension modulation of rotaxane-branched dendrimers was carried out. 2-D DOSY experiments were firstly performed to evaluate the anion-induced size change of rotaxane-branched dendrimer $G_1$. To our delight, upon the addition of TBAA, the diffusion coefficient ($D$) of rotaxane-branched dendrimer $G_1$ increased from $(7.37 \pm 0.04) \times 10^{-10}$ $m^2 s^{-1}$ to $(10.35 \pm 0.12) \times 10^{-10}$ $m^2 s^{-1}$, suggesting the decrease of its hydrodynamic size. The subsequent introduction of $NaPF_6$ into the mixture led to the

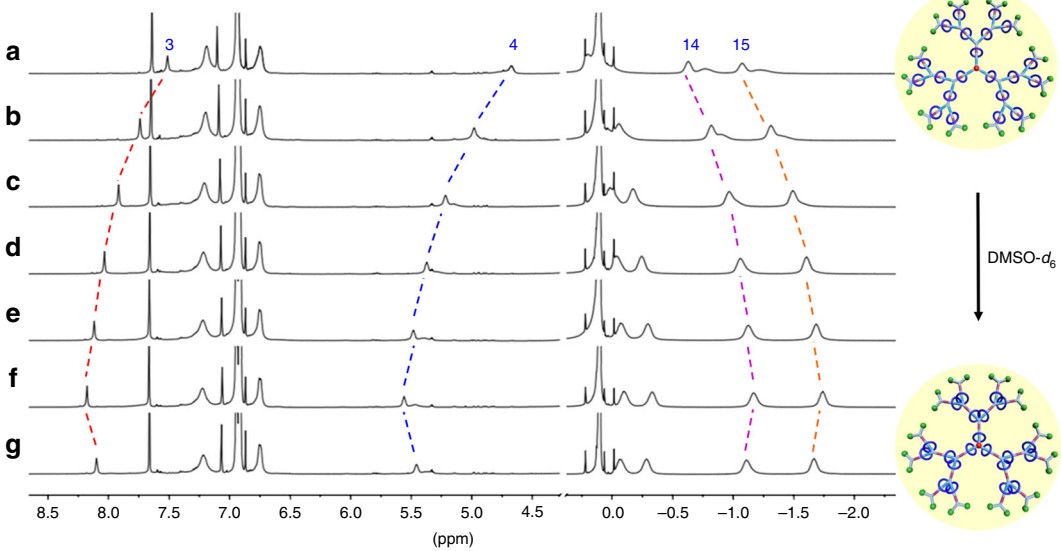

**Fig. 5** $^1$H NMR spectra (THF-$d_8$ (400 μL), 298 K, 500 MHz) of solvent-induced switching behavior of rotaxane-branched dendrimer **G₃**. **a** **G₃**; the addition of DMSO-$d_6$ in **a**: **b** 2 μL; **c** 4 μL; **d** 6 μL; **e** 8 μL; **f** 10 μL; **g** the addition of THF-$d_8$ (100 μL) in **f**

**Table 1 Diffusion coefficient (*D*) values of the anion-responsive rotaxane-branched dendrimers G₁–G₃**

|  | **G₁** | **G₁+TBAA** | **G₁+TBAA+Na⁺** |
|---|---|---|---|
| $D/10^{-10}$ m² s⁻¹ | 7.37 ± 0.04 | 10.35 ± 0.12 | 8.91 ± 0.10 |
|  | **G₂** | **G₂+TBAA** | **G₂+TBAA+Na⁺** |
| $D/10^{-10}$ m² s⁻¹ | 5.69 ± 0.06 | 8.66 ± 0.15 | 7.08 ± 0.16 |
|  | **G₃** | **G₂+TBAA** | **G₃+TBAA+Na⁺** |
| $D/10^{-10}$ m² s⁻¹ | 2.95 ± 0.07 | 5.01 ± 0.11 | 3.98 ± 0.18 |

decrease of diffusion coefficients, thus indicating the increase of the hydrodynamic size (Table 1, Supplementary Figs. 95–97). Furthermore, in order to provide further support to such stimuli-induced size modulation of rotaxane-branched dendrimer **G₁** in the solution phase, dynamic light scattering (DLS) analysis was performed. DLS investigation revealed that the size of **G₁** decreased from 2.03 to 1.63 nm with a shrinking ratio of 19.5%, which was in accord with the 2-D DOSY analysis (Supplementary Table 10 and Supplementary Fig. 118a).

In order to confirm that the size modulation of the integrated rotaxane-branched dendrimer was attributed to the rotaxane switching on each branch, the model first-generation dendrimers either without urea moiety (**G₁-a**) or without macrocycles moiety (**G₁-b**) were synthesized by employing the same controllable divergent approach from the corresponding model complexes **2-a** or **2-b**, respectively (Supplementary Figs. 65 and 66). Both model dendrimers were well-characterized by multinuclear NMR ($^1$H, $^{13}$C, and $^{31}$P) and MS analysis. In the case of the model dendrimer **G₁-a** without urea moiety, due to the absence of binding site, no obvious change was found in the $^1$H NMR spectra upon the addition of either DMSO molecule or acetate anion as stimulus (Supplementary Fig. 98). While in the case of model rotaxane dendrimer **G₁-b** without macrocycles moiety, upon the addition of DMSO molecule or acetate anion as stimulus, the existence of hydrogen bonding interactions between the urea moiety with either DMSO molecule or acetate anions was confirmed as evidenced by the remarkable chemical shifts of H$_3$ and H$_4$ in the $^1$H NMR spectrum (Supplementary Fig. 99). Since the acetate anion is a better stimulus than DMSO molecule as demonstrated by the larger downfield shifts in the $^1$H NMR

spectrum, acetate anion was selected as an external stimulus to study the size modulation property of model dendrimers by using the 2-D DOSY technique. It was found that, for both model dendrimers, almost no change of the diffusion coefficient value before and after the addition of acetate anion was observed (for **G₁-a**, $D = (13.11 ± 0.07) × 10^{-10}$ m² s⁻¹, for the mixture of **G₁-a** and TBAA, $D = (13.26 ± 0.08) × 10^{-10}$ m² s⁻¹; for **G₁-b**, $D = (9.83 ± 0.06) × 10^{-10}$ m² s⁻¹, for the mixture of **G₁-b** and TBAA, $D = (9.72 ± 0.05) × 10^{-10}$ m² s⁻¹) (Supplementary Figs. 100–105). Moreover, the DLS measurement revealed that the sizes of both model dendrimers almost maintained before and after adding 5.0 eq. of TBAA (for **G₁-a**, before: 1.35 nm, after: 1.39 nm; for **G₁-b**, before: 1.55 nm, after: 1.53 nm) as shown in Supplementary Fig. 106. These observations clearly demonstrated that the size of the model dendrimers did not change with the addition of anion, which might exclude the anion effect that caused the swelling/de-swelling of the rotaxane-branched dendrimers in this study.

Based on the aforementioned size modulation of rotaxane-branched dendrimer **G₁** attributed to the anion-induced switching, the anion-triggered switching of higher-generation rotaxane-branched dendrimers **G₂** and **G₃** was investigated. Interestingly, according to $^1$H NMR titration experiments, in these two cases, upon the addition of TBAA, two sets of peaks (for **G₂**, H$_3$: 11.49 and 10.23 ppm; H$_4$: 9.08 and 8.17 ppm; for **G₃**, H$_3$: 11.39 and 10.17 ppm; H$_4$: 8.97 and 8.11 ppm) attributed to each proton of urea moiety were observed, which might be due to the inequivalence of different generations in higher-generation rotaxane-branched dendrimers (Supplementary Table 7 and Supplementary Fig. 109 for **G₂**; Fig. 6, Supplementary Table 9 and Supplementary Fig. 112 for **G₃**). Notably, at least 10.0 equiv. of NaPF$_6$ for each urea moiety was needed to totally remove the acetate anions because of the existence of remarkable steric hindrance. Moreover, anion-induced switching of rotaxane-branched dendrimers **G₂** and **G₃** were reversible, thus allowing for the controllable switching motions of DEP5 rings on branches. Notably, in order to evaluate the cycling ability of targeted rotaxane-branched dendrimers, recycling experiments were performed by filtrating the in situ formed NaOAc precipitate through filter syringe after each cycle. As indicated by the chemical shift of proton H$_4$, in all the cases, after four full operation cycles, the chemical shift of H$_4$ could almost go back to

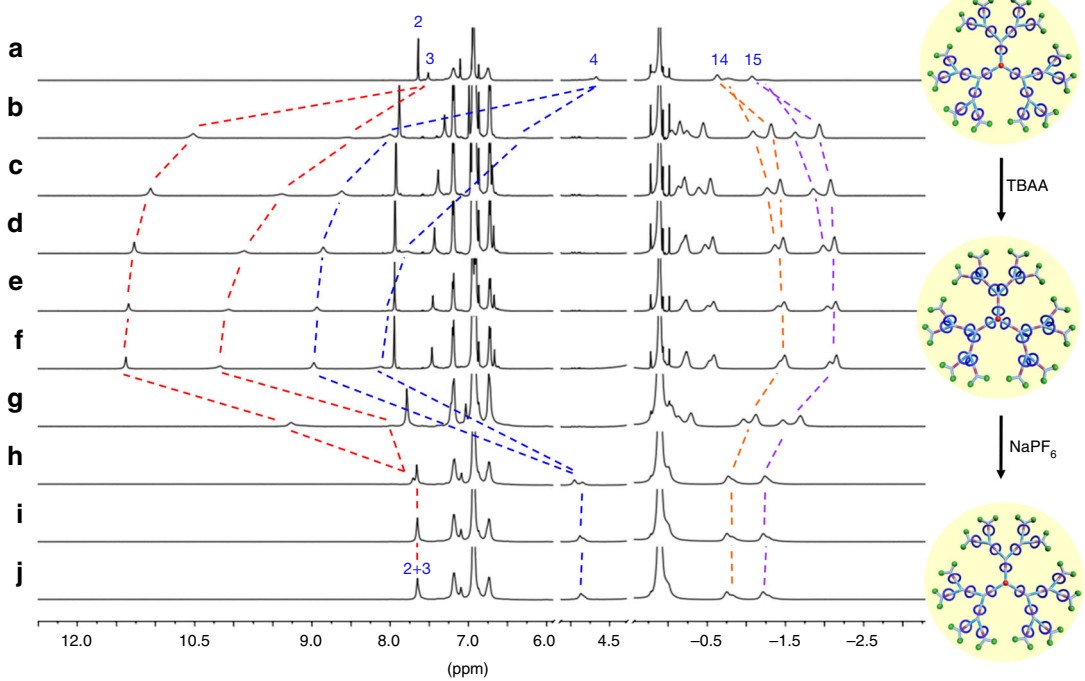

**Fig. 6** $^1$H NMR spectra (THF-$d_8$, 298 K, 500 MHz) of anion-induced switching motion of rotaxane-branched dendrimer **G$_3$**. **a** **G$_3$**; the mixture of **G$_3$** and TBAA, for each rotaxane unit: **b** TBAA (1 equiv); **c** TBAA (2 equiv); **d** TBAA (3 equiv); **e** TBAA (4 equiv); and **f** TBAA (5 equiv); and the mixture obtained after adding NaPF$_6$ to the solution in **f**, for each rotaxane unit: **g** NaPF$_6$ (5 equiv); **h** NaPF$_6$ (7 equiv); **i** NaPF$_6$ (8 equiv); **j** NaPF$_6$ (10 equiv)

the original value (Supplementary Fig. 113), thus indicating the good recycling ability of these systems.

As expected, upon the addition of acetate anions as stimulus, the reversible size modulation of **G$_2$** and **G$_3$** were evidenced by 2-D DOSY measurement (Table 1). Notably, although the diffusion coefficients could not return to the original values possibly due to the existence of the in situ formed NaOAc precipitate, the trend of the size switching was reasonable. More importantly, the exact value changes of diffusion coefficients indicated that, along with the generation increase of dendrimers, the size switching of rotaxane-branched dendrimers became more remarkable. According to the Stokes–Einstein equation, the diffusion coefficient ($D$) is directly proportional with hydrodynamic radii ($R_h$). Thus, the shrinking ratio of $R_h$ for different rotaxane dendrimers was calculated (for **G$_1$**, 29.2%; for **G$_2$**, 35.4%; for **G$_3$**, 42.5%) (Supplementary Figs.114–117). Such difference of shrinking ratio of $R_h$ might derive from the integration of multiple switchable rotaxane moieties in a monodispersed macromolecule, which can amplify the responsiveness to lead to different degree of concentration and extension. Furthermore, DLS investigation indicated that the size of **G$_2$** decreased from 3.14 to 2.28 nm with a shrinking ratio of 27.4%, and the size of **G$_3$** decreased from 4.51 to 2.75 nm with a shrinking ratio of 39.0% (Supplementary Fig. 118b, c). Both the trend of anion-induced size modulation and the shrinking ratio were in accord with the 2-D DOSY analysis, thus again confirming the anion-induced size modulation behaviors. Moreover, upon adding sodium cations to remove the acetate anions, the sizes of all rotaxane-branched dendrimers increased, thus indicating the reversibility of such size modulation processes. Similar with 2-D DOSY analysis, the sizes of the rotaxane-branched dendrimers could not fully go back to the original state, which might be caused by the existence of the in situ formed NaOAc precipitate.

In order to get insights into the morphology change of rotaxane-branched dendrimers after the addition of acetate anions as external stimulus, AFM analysis was carried out to study the morphology change before and after the addition of acetate anion. Upon the addition of TBAA, the average height of all three rotaxane-branched dendrimers displayed the obvious shrinking, with the values decreasing from 1.76 ± 0.26 to 1.52 ± 0.29 nm for **G$_1$**, from 2.70 ± 0.29 to 1.85 ± 0.25 nm for **G$_2$**, and even from 3.21 ± 0.34 to 2.05 ± 0.30 nm for **G$_3$** (Fig. 7 and Supplementary Figs. 119–121).

During the process of such anion-induced size switching of rotaxane-branched dendrimers, the addition or remove of acetate anions reversibly changed the location of DEP5 rings on the axle, thus influencing the rigidity of all branches. When DEP5 rings were located on the urea moiety and the adjacent methylene groups, the self-folding of the branches from the larger stopper site was partially inhibited. However, in the state that DEP5 rings were located on the methylene groups far from urea moiety, such self-folding process became easier, thus resulting in the shrinking of rotaxane-branched dendrimers. Along with the dimension switching induced by anion, the microenvironment of the rotaxane-branched dendrimers changed, thus offering great opportunities to explore potential applications such as controllable capture/delivery or supramolecular catalysis, etc.

## Discussion

In conclusion, by employing a controllable divergent approach, we have demonstrated the successful synthesis of dual stimuli-responsive rotaxane-branched dendrimers up to the third generation with 21 switchable rotaxane units on branches. More importantly, due to the responsiveness of the switchable rotaxane unit inserted on each branch towards DMSO molecule or acetate anion, the resultant rotaxane-branched dendrimers displayed a reversible size switching upon the external stimuli. With the addition or removal of the external stimuli, the location change of DEP5 ring on each branch led to the alteration of the rigidity of branches, which further influenced the self-folding process and finally the size of the integrated rotaxane-branched dendrimers. According to this proof-of-concept work, the controllable

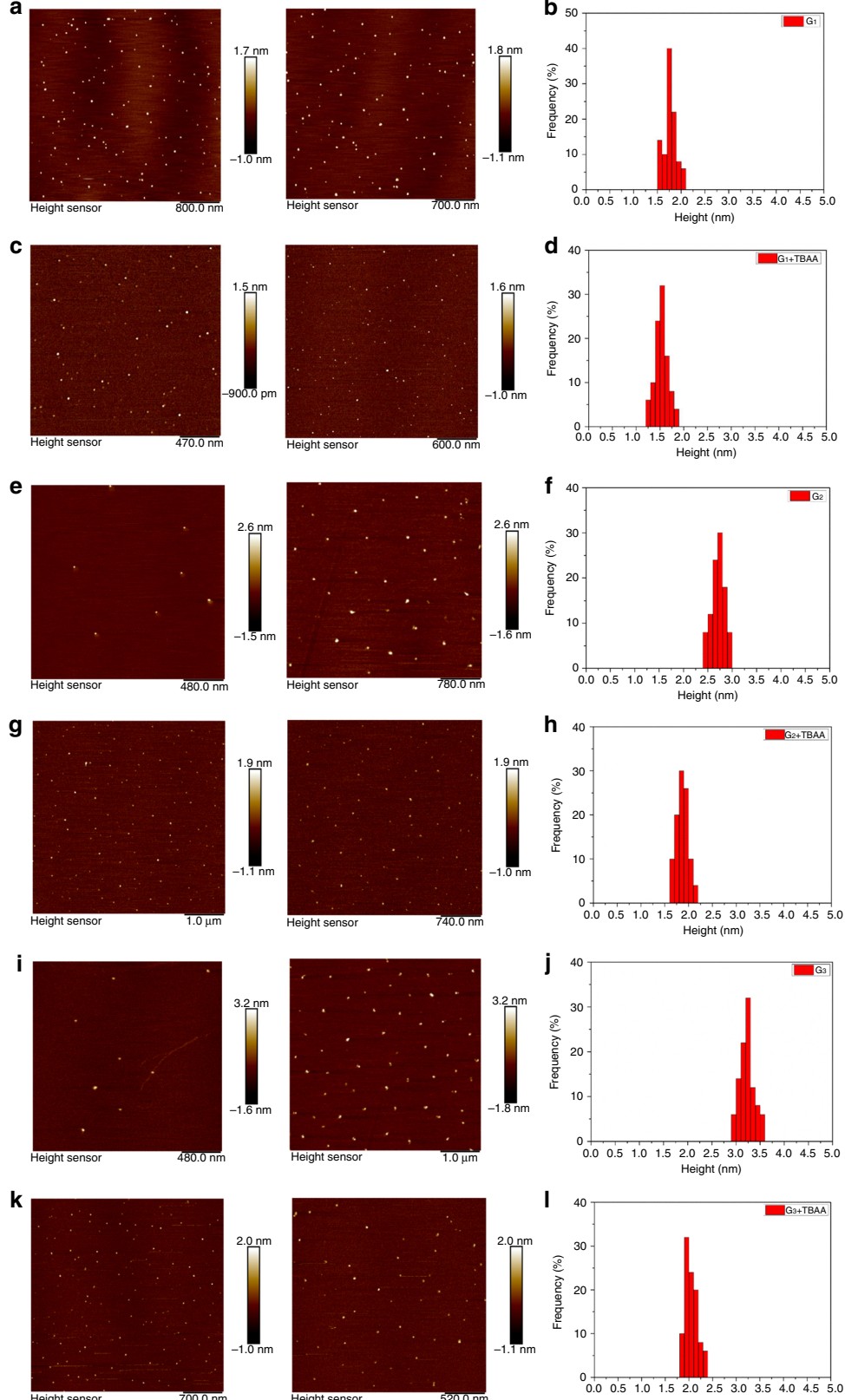

**Fig. 7** AFM images of rotaxane-branched dendrimers. **a** $G_1$; **c** $G_1$ with the addition of TBAA; **e** $G_2$; **g** $G_2$ with the addition of TBAA; **i** $G_3$; **k** $G_3$ with the addition of TBAA; and height distributions of the rotaxane-branched dendrimers **b** $G_1$, the height range is 1.76 ± 0.26 nm; **d** $G_1$ with the addition of TBAA and the height range is 1.52 ± 0.29 nm; **f** $G_2$, the height range is 2.70 ± 0.29 nm; **h** $G_2$ with the addition of TBAA and the height range is 1.85 ± 0.25 nm; **j** $G_3$, the height range is 3.21 ± 0.34 nm; **l** $G_3$ with the addition of TBAA and the height range is 2.05 ± 0.30 nm

dimension modulation of rotaxane-branched dendrimers through the integrated motions of individual rotaxane moiety has been proven to be feasible and practical, thus opening up a new avenue to the in-depth investigation of rotaxane dendrimers as dynamic functional materials.

## Methods

All solvents were dried according to standard procedures and all of them were degassed under $N_2$ for 30 min before use. All air-sensitive reactions were carried out under inert $N_2$ atmosphere. $^1H$ NMR, $^{13}C$ NMR and $^{31}P$ NMR spectra were recorded on a Bruker 300 MHz Spectrometer ($^1H$: 300 MHz; $^{31}P$: 121.4 MHz) and Bruker 400 MHz Spectrometer ($^1H$: 400 MHz; $^{13}C$: 100 MHz, $^{31}P$: 161.9 MHz) at 298 K. The $^1H$ and $^{13}C$ NMR chemical shifts are reported relative to residual solvent signals, and $^{31}P$ {$^1H$} NMR chemical shifts are referenced to an external unlocked sample of 85% $H_3PO_4$ ($\delta$ 0.0). 2-D NMR spectra ($^1H$-$^1H$ COSY, ROESY, and DOSY) were recorded on Bruker 500 MHz Spectrometer ($^1H$: 500 MHz) at 298 K. The MALDI MS experiments were carried out on a Bruker UltrafleXtreme MALDI TOF/TOF Mass Spectrometer (Bruker Daltonics, Billerica, MA), equipped with smartbeam-II laser. All spectra were measured in positive reflectron or linear mode. All the TEM measurements were performed under a Tecnai G2 20 TWIN device; the TEM samples were deposited on copper grids, followed by a slow evaporation in air at room temperature. All the AFM images were obtained on a Dimension FastScan (Bruker), using ScanAsyst mode under ambient condition. The AFM samples were prepared by drop casting method using mica sheet as substrate. UV–vis spectra were recorded in a quartz cell (light path 10 mm) on a Cary 50Bio UV-Visible spectrophotometer.

**Data availability**. The data that support the findings of this study are available from the authors on reasonable request. See author contributions for specific data sets.

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

## Acknowledgements

This work was financially supported by the NSFC/China (Nos. 21625202, 21572066, and 21672070), 973 Program (No. 2015CB856600), STCSM (No. 16XD1401000), and the Program for Changjiang Scholars and Innovative Research Team in University. We greatly appreciate Prof. Han-Yuan Gong at Beijing Normal University for his kind help with $^1$H NMR titration and Liang Zhang at The University of Manchester for helpful comments.

## Author Contributions

H.-B.Y., L.X., X.-Q.W., and W.W. conceived the project, analyzed the data, and wrote the manuscript. X.-Q.W. performed the most of experiments. W.W., W.-J.L., L.-J.C., R.Y., G.-Q.Y., Y.-X.W., Y.Z., J.H., Y.Y., H.T., and X.L. helped in experiments and data analyses. All authors discussed the results and commented on the manuscript.

## Additional information

**Competing interests:** The authors declare no competing interests.

