## [Peer Review File · Nature Communications]

Reviewers' comments:

Reviewer #1 (Remarks to the Author):

This paper discusses the synthesis characterization and switching of a few generation of rotaxane containing dendrimers. The paper is based on a previous one where a structurally similar system was made. The addition here is the switching process using DMSO and acetate anion and this will be the focus here. While the characterization of the synthesis products is acceptable (though purity not shown and PDI shows it is not completely uniform) the characterization of the switching process is not convincing. From the NMR data is clear that some motion is happening, but the nature of this motion is not clear, and if macrocycle will be minimal as it is moving slightly from one side of chain to another, i.e., the urea station is not really a station as shown by the calculation. The typical control experiments with dumbbells is missing so it is difficult to assess whether these shifts are from solvent or anion effect of real motion. The use of solvent to shift things around is not optimal as well as it limits cycles, and in this case only one is used. Precipitating the anion is also problematic as it limits cycles, and again only one is shown. The DOESY shows slight changes in size although no errors are reported so not clear what the real change it (what is the solvent effect as μ changes and what the effect of counter ion is and so on and so forth). The change in hydrodynamic area is not giving, and my estimate is that it is too small for such a system. Then comes AFM measurements that show 3 nm to 2 nm (again no errors) change in a system that has 20+ pillarenes that each have a size of roughly 1 nm!!! How is this even possible? Even if the dendrimers are squashed on the surface one would expect a larger starting height. Based on these issues this paper cannot be published in its current form

Reviewer #2 (Remarks to the Author):

Overall, the paper by Yang et al is interesting and reports novel results on the switching of dendrimer rotaxanes using either hydrogen-bonding solvents or anions to modify the structure of the target materials. In general the compounds are well characterised and the paper is convincing.

However there are a number of issues which need to be addressed. Firstly, the paper could be much more clearly written. Ironically the schemes in the supporting information are much clearer than those in the text. For example Figures S26 and S69 clearly show how the authors envisage the motion in the rotaxanes structures. The figures in the main text are far less clear, perhaps they are just too small, but at the very least the authors need to indicate what is intended by the blue and purple parts of the rotaxanes?

The NMR measurements are generally well carried out. However, I have two questions. Firstly the authors suggest in figure 4 that they are replacing the acetate anion with "Na+". This is a simplification. The authors have added NaPF₆. I am surprised that the authors haven't considered whether the PF₆⁻ anion hydrogen bonds to the urea group. Is there no indication of hydrogen bonding in this case – it is difficult to tell from Figure 4, although there do appear to be some shifts in the NMR. Have the authors used ¹⁹F NMR to investigate the PF₆⁻ anion? I am surprised that the authors have not evaluated the binding constants for the various stimuli, particularly the acetate. They seem to have the relevant information in the NMR spectra. Does this binding constant vary depending on the generation of the dendrimer?

On page 5 the authors state that TEM and AFM provide "a visual proof" – this is an incorrect statement – electron microscopy and AFM do not provide visual proof. They also state that the samples are shown to be near spherical – where is the evidence for his statement? The AFM just gives height profile and does not indicate spherical shape – AFM cannot measure this. Similarly these TEM measurements do not indicate spherical shapes, have the authors performed 3D imaging? I think not.

The TEM image shown in Figure S61 is rather unconvincing. The authors highlight two features which correspond to their ideal view of the dendrimer but there are also a number of features which appear to be of different sizes. What are these? The authors need to give an indication of the spread of sizes for the TEM images indicating how the size of the dendrimers varies in the sample.

For the anion containing dendrimers, are the authors sure that the anion is still bound following deposition of the samples? The authors do not appear to state what substrate the samples were deposited onto for the AFM measurements and how they were deposited. This needs to be rectified.

Figure 6 is unclear, the height profiles are fuzzy, perhaps this is a pdf conversion problem.

In the G3 dendrimer the "solvent/anion switched" form seems very congested from the schematic representation shown in Figure S79. Of course, this is only a scheme but it does raise the question of steric congestion for these large, branched systems. Is there any indication of whether there are interactions between pillararenes on adjacent branches? The authors should look at NOESY NMR of these dendrimer species to evaluate inter-branch interactions.

Overall, the paper is interesting but needs additional experiments and evaluation prior to any possible publication.

Reviewer #3 (Remarks to the Author):

Review for manuscript NCOMMS-17-31552

"Dual Stimuli-responsive Rotaxane-branched Dendrimers with Reversible Dimension Modulation"
The manuscript by Yang and co-workers build a series poly-[2]-rotaxane containing organometallic dendrimers and examine the increase and decrease of the size of the dendrimers when treated with dimethylsulfoxide (DMSO) molecules or acetate anions. Initially, they report the synthesis and switching behaviour of the "simple" organometallic [2]rotaxane. The system is well characterised using NMR, MS and calculations. It would be nice if the authors provided the energies of the calculated structures but from the NMR data it is clear that the switching occurs. The authors then use the organometallic [2]rotaxane as a building block to generate the branched dendrimers, using a divergent approach allowed the successful synthesis of a new family of rotaxane-branched dendrimers up to a third generation system which contained twenty-one switchable rotaxane moieties.

Again the dendrimers were well characterised using NMR, MS, GPC and AFM, TEM was less convincing, but DOSY NMR also confirmed the formation and size of the systems. The authors then repeated the switching experiments and showed that the addition and removal of DMSO or acetate anions resulted in switching of the macrocycle along the rotaxane thread (as observed in the simpler material). They also found that the size of the dendrimers changes when treated with the external stimuli, and suggest that the movement of the macrocycle is amplified in the dendrimers generating the expansion and contraction of the dendrimers. This could be correct but the authors have not done the controls that would confirm their hypothesis. The expansion and contraction (swelling/deswelling) of polymer, gels and dendrimers is known to occur when the solvent or anions are changed thus the observed size change may just be due to the swelling/deswelling of the dendrimers and not connected to the movement of the macrocycle. To prove/disprove this the authors need to make the series of analogues dendrimers that do not feature the macrocycle and examine the switching and the size change. This must be done before publication as the current work does not provide enough evidence that the change in size of the system is related to the

switching of the rotaxanes.

Additionally, some of the English need rewording and the some of the pictures also need to fixed! The resolution of Scheme 1 and Fig.1 is poor and need to be fixed. The X axis on the NMR spectra are impossible to read and for some reason seem to say (f1) ppm rather than the correction delta (ppm)

The resolution of 5(a) and 6(a-f) need to be improve the blue lines in 6 should be made thicker.

The histogram in figure 5 is not clear what are all the different colors referring to?

The authors should also state somewhere that addition of Na⁺ in THF causes the precipitation of NaOAc (s) in order to remove the OAc anions from solution as this was not immediately clear in the manuscript

This is certainly nice interesting work and it could be great paper once the control experiments are done.

Therefore, I recommend that the manuscript be rejected then resubmitted after the correct controls reactions are carried out and the other more minor change are attended to.

April 23, 2018

Response to Referee 1:

This paper discusses the synthesis characterization and switching of a few generation of rotaxane containing dendrimers. The paper is based on a previous one where a structurally similar system was made. The addition here is the switching process using DMSO and acetate anion and this will be the focus here. While the characterization of the synthesis products is acceptable (though purity not shown and PDI shows it is not completely uniform) the characterization of the switching process is not convincing.

Reply: We fully understand the reviewer's concern about the purity of the resultant rotaxane-branched dendrimers with numbers of rotaxane units and large molecular weights. In the original version, in order to evaluate the purity of the targeted rotaxane-branched dendrimers, one-dimensional (1-D) NMR, two-dimensional (2-D) DOSY NMR spectroscopy, and gel permeation chromatography (GPC) measurement were performed, which provided the strong supports to the purity of the obtained rotaxane-branched dendrimers. For instance, for all three rotaxane-branched dendrimers, only one set of signals was observed in 2-D DOSY spectra, thus indicating the existence of the sole species. In the case of GPC analysis, both **G₂** and **G₃** exhibited a single peak (Fig. S48-49), respectively, again conforming the purity of rotaxane-branched dendrimers **G₂** and **G₃**.

According to the reviewer's comment, GPC measurement and element analysis of the first-generation rotaxane-branched dendrimer **G₁** were further carried out to provide the more support to the purity issue in the revised manuscript. As shown in Fig. R1, a single sharp peak with a very

narrow PDI (1.03) was observed in the GPC spectrum of **G**₁. In addition, the elemental analysis of **G**₁ agreed with the targeted rotaxane-branched dendrimer with the reasonable purity (Anal. Calcd. for [**G**₁ + 3CH₂Cl₂] C₃₅₇H₅₂₂Cl₆N₆O₃₆P₆Pt₃Si₆: C, 64.71; H, 7.94; N, 1.27, Found: C, 64.51; H, 8.32; N, 1.13).

Fig. R1 GPC spectrum of the rotaxane-branched dendrimer **G**₁.

Therefore, based on the combination of 1-D NMR (¹H and ³¹P NMR), 2-D DOSY spectroscopy, GPC measurement, and element analysis, the purity of the targeted rotaxane-branched dendrimers should be confirmed. The additional experiment results have been added in Supplementary Information.

From the NMR data is clear that some motion is happening, but the nature of this motion is not clear, and if macrocycle will be minimal as it is moving slightly from one side of chain to another, i.e., the urea station is not really a station as shown by the calculation. The typical control experiment with dumbbells is missing so it is difficult to assess whether these shifts are from solvent or anion effect of real motion.

Reply: We greatly appreciate the reviewer's suggestion to convince the switching process of rotaxane moiety. According to the reviewer's advice, a series of control experiments were carried out to obtain the additional supports for such stimuli-induced switching behaviors. Two model complexes either without the urea moiety (**2-a**) or without the pillar[5]arene macrocycle (**2-b**) were synthesized as shown in Scheme R1.

Scheme R1 The chemical structures and cartoon representation of (a) [2]rotaxane building block **2**; (b) the model complex [2]rotaxane **2-a** without urea moiety as the stimuli-responsive site; (c) the model complex platinum-acetylide building block **2-b** without pillar[5]arene as the wheel.

With these two model complexes in hand, the control experiments of stimuli-responsive behavior were performed. In the case of model complex **2-a** without urea moiety, upon the addition of DMSO- d_6 (10 μ L) or TBAA (5.0 eq.) into the solution of **2-a** in THF- d_8 (9.0 mM, 400 μ L), the resultant spectra showed no obvious change compared with the original spectrum of **2-a** as indicated in ^1H NMR spectra (Fig. R2). While for the [2]rotaxane building block **2**, upon the addition of either DMSO- d_6 (10 μ L) or TBAA (5.0 eq.) as the stimulus at the same concentration (0.4 mM), the obvious downfield shifts of the protons on urea moiety (H_3 and H_4) were observed (Fig. R3). The combination of these findings suggested that the urea moiety did act as a binding site interacting with the DMSO molecule or anion species.

More importantly, by comparing the ^1H NMR spectra of model complex **2-b** with the [2]rotaxane building block **2** before and after the addition the DMSO molecule or anion species, the translational motion of DEP5 macrocycle along the axle was confirmed. When comparing the ^1H NMR spectrum of [2]rotaxane **2** with the one of model complex **2-b** (Fig. R4b and c, Fig. R5b and c), the obvious upfield shifts of urea protons (H_3 and H_4) were observed in the ^1H NMR spectrum of [2]rotaxane **2**, suggesting the encapsulation of urea moiety and the adjacent methylene units within the cavity of DEP5. While upon the addition of DMSO molecule as stimulus, as indicated in the ^1H NMR spectra of the model complex **2-b** and the corresponding [2]rotaxane **2** (Fig. R4a and Fig. R4d, respectively), the protons on urea moiety (H_3 and H_4) downfield shifted to the almost same position. In couple with

the obvious upfield shifts of the methylene units (H_{13} , H_{14} , H_{15} , and H_{16}) in the axle of [2]rotaxane **2**, the stimuli-induced movement of DEP5 macrocycle from urea moiety to the neutral alkyl chain on the other side was confirmed. In the case of acetate anion as stimulus, the same result was observed as shown in Fig. R5. These findings strongly supported the existence of translational motion of DEP5 macrocycle along the axle stimulated by DMSO molecule or acetate anion.

Moreover, according to the calculated structure, due to the existence of stronger hydrogen bonding between urea moiety and ethoxy group of pillar[5]arene macrocycle, the urea moiety and the adjacent methylene units would coordinately serve as one station, which would be preferentially positioned within the cavity of the pillar[5]arene. Such favored complex state was further confirmed by 2-D ROESY spectrum (Fig. S23), in which the correlations between proton H_4 of urea moiety with the aromatic protons (H_a) and the bridged methylene protons (H_c) of pillar[5]arene were observed. The addition of DMSO or acetate anions as stronger hydrogen bonding acceptors would destroy the complexation between urea moiety and pillar[5]arene macrocycle, thus leading to the translational motion of pillar[5]arene ring toward to the neutral alkyl chain on the other side.

The discussion about the additional control experiments was added in main text of the revised version. In particular, to avoid the possible misunderstanding, a revision on description of the original state of [2]rotaxane building block **2** was made as follows in the revised manuscript: “*Due to the stronger hydrogen bonding interactions between the ethoxy group of pillararene macrocycle and the urea moiety compared with the $CH\cdots\pi$ interactions between pillararene macrocycle and neutral alkyl chain, the urea moiety and the adjacent methylene units would be preferentially positioned within the cavity of the pillar[5]arene*”. Moreover, the addition control experiment results have been provided in Supplementary Information.

Fig. R2 ^1H NMR spectra (THF- d_8 , 298 K, 500 MHz) of model complex **2-a** with the addition of DMSO- d_6 (10 μL) (bottom); **2-a** (middle); **2-a** with the addition of TBAA (5 eq.) (top).

Fig. R3 ^1H NMR spectra (THF- d_8 , 298 K, 500 MHz) of [2]rotaxane building block **2** with the addition of DMSO- d_6 (10 μL) (*bottom*); [2]rotaxane building block **2** (*middle*); **2** with the addition of TBAA (5.0 eq.) (*top*).

Fig. R4 ^1H NMR spectra (THF- d_8 , 298 K, 500 MHz) of a) [2]rotaxane building block **2** with the addition of DMSO- d_6 (10 μL); b) [2]rotaxane building block **2**; c) model complex **2-b**; d) model complex **2-b** with the addition of DMSO- d_6 (10 μL);

Fig. R5 ^1H NMR spectra (THF- d_8 , 298 K, 500 MHz) of a) [2]rotaxane building block **2** with the addition of TBAA (5.0 eq.); b) [2]rotaxane building block **2**; c) model complex **2-b**; d) model complex **2-b** with the addition of TBAA (5.0 eq.).

The use of solvent to shift things around is not optimal as well as it limits cycles, and in this case only one is used. Precipitating the anion is also problematic as it limits cycles, and again only one is shown.

Reply: We fully understand the reviewer's concern about the cycle issue of the switchable rotaxane-branched dendrimers using solvent or anion as stimulus. It is true that the addition of DMSO molecule as stimulus changed the concentration of the whole system. Moreover, it is hard to remove the DMSO solvent from the system, thus hampering the recycling process. While the cycling experiment of [2]rotaxane building block **2** as well as the rotaxane-branched dendrimers **G1-G3** could be performed by adding acetate anion as stimulus. In order to exclude the effect of the *in-situ* formed NaOAc precipitate, the filtration of NaOAc through filter syringe was performed after each cycle. As the specific proton in the urea moiety, proton H_4 was selected to investigate the chemical shift upon cycling. As shown in Fig. R6, in all the cases, after four full operation cycles, the chemical shift of H_4 could almost go back to their original value, thus indicating the good recycling ability of these systems.

The discussion about the additional recycling experiments was added in main text of the revised version. Moreover, the recycling experiment results have been provided in Supplementary Information.

Fig. R6 Recycling experiments of anion-induced switching of [2]rotaxane building block **2** and rotaxane-branched dendrimers **G₁-G₃**.

The DOESY shows slight changes in size although no errors are reported so not clear what the real change it (what is the solvent effect as μ changes and what the effect of counter ion is and so on and so forth). The change in hydrodynamic area is not giving, and my estimate is that it is too small for such a system.

Reply: Based on the reviewer's suggestion, the updated results of diffusion coefficients (D) with the errors were provided as listed below (Table R1).

Table R1. Diffusion coefficient (D) of rotaxane-branched dendrimers **G₁-G₃** with the addition and removal of TBAA.

	G₁	G₁+TBAA	G₁+TBAA+Na⁺
$D/10^{-10} \text{m}^2 \text{s}^{-1}$	7.37 ± 0.04	10.35 ± 0.12	8.91 ± 0.10
	G₂	G₂+TBAA	G₂+TBAA+Na⁺
$D/10^{-10} \text{m}^2 \text{s}^{-1}$	5.69 ± 0.06	8.66 ± 0.15	7.08 ± 0.16
	G₃	G₃+TBAA	G₃+TBAA+Na⁺
$D/10^{-10} \text{m}^2 \text{s}^{-1}$	2.95 ± 0.07	5.01 ± 0.11	3.98 ± 0.18

According to above-listed values, the obvious changes of the diffusion coefficients (D) of rotaxane-branched dendrimers were observed with the addition of TBAA. It should be noted that the

changes of D value induced by anion stimulus were beyond the error range, thus indicating the existence of the hydrodynamic size modulations upon the external stimulus. Actually, the employment of DOSY technique has been a generally-accepted and widely-used method to evaluate the size change of dendrimers by comparing the value of diffusion coefficients (*Macromolecules* **1994**, *27*, 3464; *Macromolecules* **2001**, *34*, 1797; *Angew. Chem. Int. Ed.*, **2005**, *44*, 1053; *Chem. Soc. Rev.*, **2008**, *37*, 479; *Macromolecules* **2010**, *43*, 9248; *Macromol. Chem. Phys.* **2005**, *206*, 1288; etc.).

In addition, in order to exclude the effect of the counter ion in the 2-D DOSY analysis, two model first-generation dendrimers either without urea moiety (**G1-a**) or without DEP5 macrocycle (**G1-b**) were synthesized by employing the same controllable divergent approach from the corresponding model complexes **2-a** or **2-b**, respectively (Scheme R2 and R3). In both cases, upon the addition of the excess TBAA, almost no change of diffusion coefficients (D) was observed (For **G1-a**, $D = (13.11 \pm 0.07) \times 10^{-10} \text{ m}^2/\text{s}$, for the mixture of **G1-a** and TBAA, $D = (13.26 \pm 0.08) \times 10^{-10} \text{ m}^2/\text{s}$; for **G1-b**, $D = (9.83 \pm 0.06) \times 10^{-10} \text{ m}^2/\text{s}$, for the mixture of **G1-b** and TBAA, $D = (9.72 \pm 0.05) \times 10^{-10} \text{ m}^2/\text{s}$) as shown in Fig. R7-10, which might rule out the effect of the counter ion in this study.

Scheme R2. The synthesis route of model first-generation rotaxane dendrimer **G1-a** from the corresponding building block **2-a**.

Scheme R3. The synthesis route of model first-generation platinum-acetylide dendrimer **G_{1-b}** from the corresponding building block **2-b**.

Fig. R7 2-D DOSY spectrum (THF-*d*₈, 298 K, 500 MHz) of model rotaxane dendrimer **G_{1-a}**.

Fig. R8 2-D DOSY spectrum (THF-*d*₈, 298 K, 500 MHz) of model rotaxane dendrimer **G1-a** with the addition of TBAA (15.0 eq.).

Fig. R9 2-D DOSY spectrum (THF-*d*₈, 298 K, 500 MHz) of model platinum-acetylide dendrimer **G1-b**.

Fig. R10 2-D DOSY spectrum (THF- d_8 , 298 K, 500 MHz) of model platinum-acetylide dendrimer **G1-b** with the addition of TBAA (15.0 eq.).

Notably, in order to obtain the further support to the stimuli-induced size modulation of rotaxane-branched dendrimers in solution phase, the additional dynamic light scattering (DLS) analysis was performed. To our delight, according to the DLS results, anion-induced size shrinking of all three rotaxane-branched dendrimers **G1-G3** were confirmed as listed in Table R2 and Fig. R11. For example, in the case of **G1**, the DLS investigation revealed that the size decreased from 2.03 nm to 1.63 nm with a shrinking ratio of 19.5%, for **G2** from 3.14 nm to 2.28 nm with a shrinking ratio of 27.4%, and for **G3** from 4.51 nm to 2.75 nm with a shrinking ratio of 39.0%. More importantly, both the trend of anion-induced size modulation and the shrinking ratio were in accord with the DOSY analysis, thus again demonstrating the anion-induced size modulation behaviors. Moreover, upon adding sodium cations to remove the acetate anions, the sizes of all rotaxane-branched dendrimers increased, thus indicating the reversibility of such size modulation processes.

The discussion about the additional control experiments and DLS investigation was added in main text of the revised version. Moreover, the results of the control experiments and DLS study have been provided in Supplementary Information.

Table R2. DLS data of anion-induced size switching of rotaxane-branched dendrimers **G₁-G₃**.

	G₁	G₁+TBAA	G₁+TBAA+Na⁺
Size (d.nm)	2.03	1.63	2.01
	G₂	G₂+TBAA	G₂+TBAA+Na⁺
Size (d.nm)	3.14	2.28	2.81
	G₃	G₃+TBAA	G₃+TBAA+Na⁺
Size (d.nm)	4.51	2.75	3.79

Fig. R11 DLS spectra of (a) rotaxane-branched dendrimers **G₁-G₃**; (b) anion-induced size switching of **G₁**; (c) anion-induced size switching of **G₂**; (d) anion-induced size switching of **G₃**.

Then comes AFM measurements that show 3 nm to 2 nm (again no errors) change in a system that has 20+ pillarenes that each have a size of roughly 1 nm!!! How is this even possible? Even if the dendrimers are squashed on the surface one would expect a larger starting height. Based on these issues this paper cannot be published in its current form.

Reply: According to the reviewer's suggestion, AFM analysis of more than 50 samples on the surface was performed for each rotaxane-branched dendrimer. The height ranges of all rotaxane-branched dendrimers **G₁-G₃** before and after the addition of acetate anions were evaluated (Fig. R12-R17). According to the resultant height information, the updated height values with errors were

presented in the revised manuscript (*Before the addition of acetate anions*: for **G₁**, 1.76 ± 0.26 nm; for **G₂**, 2.70 ± 0.29 nm; for **G₃**, 3.21 ± 0.34 nm. *After the addition of acetate anions*: for **G₁**, 1.52 ± 0.29 nm; for **G₂**, 1.85 ± 0.25 nm; for **G₃**, 2.05 ± 0.30 nm.).

We fully understand the reviewer's concern on the relatively small heights of such huge rotaxane-branched dendrimers. As we know, AFM analysis usually provides the height information of dendrimers on the surface. Different from the fully expanded conformation in the solution state, when depositing the dendrimer samples on the substrate, the surface-induced de-conformation and/or structural collapse by the solvent loss will lead to the observation that the measured height of dendrimers is much smaller than their ideal-sphere diameter. For instance, in the case of classical dendrimer PAMAM, AFM analysis indicated that the height of the eighth generation PAMAM with thousands of repeated units on a naked Au surface ranged from 3.5 to 4.0 nm, which was about 60% less than its ideal-sphere diameter (*J. Am. Chem. Soc.* **1998**, *120*, 5323). In this study, although there are twenty-one branches in the dendrimer skeleton of **G₃**, these rotaxane moieties are not fully extended and standing right on the surface. This might be the reason why its height is relatively small. Furthermore, according to our previous report (*Proc. Natl. Acad. Sci.* **2015**, *112*, 5597), the height of an analogue rotaxane dendrimer with the similar skeleton was about 3.3 nm, which could serve as another support for the reliability of the height information in this study.

The discussion about the additional AFM experiments was added in main text of the revised version. Moreover, the updated AFM results have been provided in Supplementary Information.

Fig. R12 (a-b) AFM images and (c) height distributions of the rotaxane-branched dendrimer **G₁**. The height range is 1.76 ± 0.26 nm.

Fig. R13 (a-b) AFM images and (c) height distributions of the rotaxane-branched dendrimer **G₂**. The height range is 2.70 ± 0.29 nm

Fig. R14 (a-b) AFM images and (c) height distributions of the rotaxane-branched dendrimer **G₃**. The height range is 3.21 ± 0.34 nm.

Fig. R15 (a-b) AFM images and (c) height distributions of rotaxane-branched dendrimer **G₁** with the addition of TBAA (5.0 eq. for each urea unit). The height range is 1.52 ± 0.29 nm.

Fig. R16 (a-b) AFM images and (c) height distributions of rotaxane-branched dendrimer **G₂** with the addition of TBAA (5.0 eq. for each urea unit). The height range is 1.85 ± 0.25 nm.

Fig. R17 (a-b) AFM images and (c) height distributions of rotaxane-branched dendrimer G_3 with the addition of TBAA (5.0 eq. for each urea unit). The height range is 2.05 ± 0.30 nm.

Reviewer #2 (Remarks to the Author):

Overall, the paper by Yang et al is interesting and reports novel results on the switching of dendrimer rotaxanes using either hydrogen-bonding solvents or anions to modify the structure of the target materials. In general the compounds are well characterised and the paper is convincing.

However there are a number of issues which need to be addressed. Firstly, the paper could be much more clearly written. Ironically the schemes in the supporting information are much clearer than those in the text. For example Figures S26 and S69 clearly show how the authors envisage the motion in the rotaxanes structures. The figures in the main text are far less clear, perhaps they are just too small, but at the very least the authors need to indicate what is intended by the blue and purple parts of the rotaxanes?

Reply: According to the reviewer's advice, the English writing has been re-organized and well polished throughout the whole manuscript. In addition, during the doc-pdf file conversion process of the first round submission, most of figures became fuzzy. In the revised manuscript, all figures were re-edited and submitted in high-resolution form. For instance, in the revised Scheme 2, two binding sites in the rotaxane structure were highlighted in blue and purple, respectively, which were in accord with the cartoon representation.

The NMR measurements are generally well carried out. However, I have two questions. Firstly, the authors suggest in figure 4 that they are replacing the acetate anion with "Na⁺". This is a simplification. The authors have added NaPF₆. I am surprised that the authors haven't considered whether the PF₆⁻ anion hydrogen bonds to the urea group. Is there no indication of hydrogen bonding in this case – it is difficult to tell from Figure 4, although there do appear to be some shifts in the NMR. Have the authors used ¹⁹F NMR to investigate the PF₆⁻ anion?

Reply: In principle, as a weak counter anion, the hydrogen bonding interaction between PF_6^- and the urea group is much weaker compared with that of acetate anion. According to the reviewer's suggestion, to evaluate the influence of PF_6^- anions to the anion-induced switching of rotaxane moiety, a control experiment was performed, in which the excess amounts of NaPF_6 (5.0 eq.) were sequentially added into the $\text{THF-}d_8$ solution of [2]rotaxane **2** and ^1H and ^{19}F NMR spectra were recorded. As shown in ^1H NMR (Fig. R18) spectrum, with the addition of PF_6^- , only protons H_3 and H_4 ascribed to the urea moiety and protons H_{14} and H_{15} displayed very slight shifts, while other peaks stayed almost unchanged. In addition, almost no changes was observed in ^{19}F NMR spectra, indicating very weak hydrogen bonding interactions between PF_6^- anion and the urea moiety.

Furthermore, the competitive binding experiments were carried out. Firstly, 5.0 eq. acetate anion was added into the $\text{THF-}d_8$ solution of [2]rotaxane **2**, which led to the remarkably shifts of the corresponding protons H_3 and H_4 . Then 5.0 eq. TBAPF_6 was added into the above-mentioned solution. No obvious change in ^1H NMR spectrum was observed (Fig. R19), thus indicating very weak binding ability of PF_6^- towards the urea moiety compared with the acetate anions. Thus, the influence of PF_6^- to the anion-induced switching of rotaxane-branched dendrimers could be excluded in this study.

The discussion about the additional control experiment and competitive experiment was added in main text of the revised version. Moreover, the results of the additional control experiment and competitive experiment have been provided in Supplementary Information.

Fig. R18 Left: ^1H NMR spectra ($\text{THF-}d_8$, 298 K, 500 MHz) of a) [2]rotaxane **2**; the mixture of **2** and NaPF_6 : b) NaPF_6 (1.0 eq.); (c) NaPF_6 (2.0 eq.); d) NaPF_6 (3.0 eq.); e) NaPF_6 (4.0 eq.); f) NaPF_6 (5.0 eq.). Right: ^{19}F NMR spectra ($\text{THF-}d_8$, 298 K, 500 MHz) of a) NaPF_6 ; the mixture of **2** and NaPF_6 : b) NaPF_6 (1.0 eq.); (c) NaPF_6 (2.0 eq.); d) NaPF_6 (3.0 eq.); e) NaPF_6 (4.0 eq.); f) NaPF_6 (5.0 eq.).

Fig. R19 ^1H NMR spectra ($\text{THF-}d_8$, 298 K, 500 MHz) of [2]rotaxane **2** (*bottom*); **2** with the addition of TBAA (5 eq.) (*middle*); the mixture obtained after adding TBAPF_6 to the solution in middle (*top*).

I am surprised that the authors have not evaluated the binding constants for the various stimuli, particularly the acetate. They seem to have the relevant information in the NMR spectra. Does this binding constant vary depending on the generation of the dendrimer?

Reply: According to the reviewer's suggestion, the acetate binding affinities of [2]rotaxane building block **2** and first-generation rotaxane-branched dendrimer **G**₁ were determined by ^1H NMR titrations with acetate anion (TBAA) in $\text{THF-}d_8$. In the case of [2]rotaxane **2**, the data was fitted to a 2 : 3 binding model (**2**: acetate anion) as confirmed by Job plot analysis (Fig. R20a) and the anion binding constants were calculated to be $\log K_1 = 4.74 \pm 0.2$, $\log K_2 = 4.01 \pm 0.2$, and $\log K_3 = 2.74 \pm 0.5$ (Table 3 and Fig. R20b). For the rotaxane-branched dendrimer **G**₁, Job plot analysis indicated a 1:3 (**G**₁: acetate anion) binding stoichiometry (Fig. R21a), and the binding constants for acetate anion were calculated to be $\log K_1 = 4.19 \pm 0.2$, $\log K_2 = 3.43 \pm 0.2$, $\log K_3 = 3.21 \pm 0.2$ (Table R3 and Fig. R21b).

It should be noted that the higher generation rotaxane-branched dendrimers **G**₂ and **G**₃ possessed nine and twenty-one urea moieties, respectively. It is too complicated to calculate the binding constants. As indicated in ^1H NMR spectra of rotaxane-branched dendrimers after the addition of anion species (Fig. S74-75), the binding behavior of the urea moieties on different branches was different. For instance, upon the addition of TBAA to **G**₂ and **G**₃, respectively, two sets of peaks attributed to each proton of urea moiety were observed (for **G**₂, H₃: 11.49 and 10.23 ppm; H₄: 9.08 and 8.17 ppm; for **G**₃, H₃: 11.39 and 10.17 ppm, H₄: 8.97 and 8.11 ppm). According to the peak splitting of urea moieties upon the addition of external stimulus, the binding behavior of rotaxane-branched dendrimers towards acetate anion did display generation-dependent feature in this study.

The discussion about the additional ^1H NMR titration experiment was added in main text of the revised version. Moreover, the results of the additional ^1H NMR titration experiment have been provided in Supplementary Information.

Table R3. Binding constants of [2]rotaxane **2** and rotaxane-branched dendrimer **G₁**.

	$\log K_1$	$\log K_2$	$\log K_3$
2	4.74 ± 0.2	4.01 ± 0.2	2.74 ± 0.5
G₁	4.19 ± 0.2	3.43 ± 0.2	3.21 ± 0.2

Fig. R20 (a) Job plot for [2]rotaxane **2**-acetate anion complex in THF- d_8 ($[\mathbf{2}] + [\text{anion}] = 4 \text{ mM}$); (b) The ^1H NMR titration isotherm of [2]rotaxane **2** with the addition of acetate anion (TBAA) recorded at 500 MHz in THF- d_8 at 298 K.

Fig. R21 (a) Job plot for rotaxane-branched dendrimer **G₁**-acetate anion complex in THF- d_8 ($[\mathbf{G}_1] + [\text{anion}] = 4 \text{ mM}$); (b) The ^1H NMR titration isotherm of rotaxane-branched dendrimer **G₁** with the addition of acetate anion (TBAA) recorded at 500 MHz in THF- d_8 at 298 K.

On page 5 the authors state that TEM and AFM provide “a visual proof” – this is an incorrect statement – electron microscopy and AFM do not provide visual proof. They also state that the samples are shown to be near spherical – where is the evidence for his statement? The AFM just gives height profile and does not indicate spherical shape – AFM cannot measure this. Similarly these TEM measurements do not indicate spherical shapes, have the authors performed 3D imaging? I think not.

Reply: We agree with the points raised by the reviewer about the TEM and AFM analysis. According to the reviewer’s suggestion, the 3-D imaging of the rotaxane-branched dendrimers was performed in TEM analysis. Unfortunately, due to the relative small sizes of these rotaxane-branched dendrimers, it was hard to focus the images when varying the angles during the TEM measurement. So far, no satisfied 3-D image of TEM was obtained. Based on the reviewer’s suggestion, “a visual proof” was changed to “another proof” to avoid the possible misunderstanding in the revised manuscript. In addition, the shape description of the samples as “near-spherical” was deleted.

The TEM image shown in Figure S61 is rather unconvincing. The authors highlight two features which correspond to their ideal view of the dendrimer but there are also a number of features which appear to be of different sizes. What are these? The authors need to give an indication of the spread of sizes for the TEM images indicating how the size of the dendrimers varies in the sample.

Reply: According to the reviewer’s suggestion, a size range for each rotaxane-branched dendrimer was obtained by measuring about 50 samples in different TEM images as shown in Fig. R22-24 (for **G**₁, 1.86 ± 0.24 nm; for **G**₂, 2.60 ± 0.27 nm; for **G**₃, 3.55 ± 0.32 nm). As shown in TEM images of **G**₂ in Figure S61 in the original version, most rotaxane-branched dendrimers featured almost the similar sizes. However, due to different adsorption geometry and de-conformation of the samples on the substrate, the size of rotaxane-branched dendrimers revealed by TEM analysis might be slightly different.

Fig. R22 (a-c) TEM images and (d) contour length distributions of the rotaxane-branched dendrimer **G**₁. The length range is 1.86 ± 0.24 nm.

Fig. R23 (a-c) TEM images and (d) contour length distributions of the rotaxane-branched dendrimer **G2**. The length range is 2.60 ± 0.27 nm.

Fig. R24 (a-c) TEM images and (d) contour length distributions of the rotaxane-branched dendrimer **G3**. The length range is 3.55 ± 0.32 nm .

The discussion about the additional TEM experiments was added in main text of the revised version. Moreover, the updated TEM results have been provided in Supplementary Information.

For the anion containing dendrimers, are the authors sure that the anion is still bound following deposition of the samples? The authors do not appear to state what substrate the samples were deposited onto for the AFM measurements and how they were deposited. This needs to be rectified.

Reply: We fully understand the reviewer's concern about AFM measurement. In this study, the sample for AFM measurement was prepared according to the following procedure. The solution of anion-complexed rotaxane-branched dendrimers in THF was diluted to 10^{-7} M. Then the sample for AFM measurement was prepared by drop casting method using mica sheet as substrate. In principle, both the dilution and deposition process will not destroy the complexation between anions and urea moieties because there is no external factor disturbing the hydrogen bonding interactions upon diluting and depositing.

According to the reviewer's suggestion, a brief description on AFM measurement has been added in the Methods part in the text.

Figure 6 is unclear, the height profiles are fuzzy, perhaps this is a pdf conversion problem.

Reply: Indeed, during the doc-pdf file conversion process of the first round submission, most of figures became fuzzy. In the revised manuscript, all figures were submitted in high-resolution form.

In the G3 dendrimer the “solvent/anion switched” form seems very congested from the schematic representation shown in Figure S79. Of course, this is only a scheme but it does raise the question of steric congestion for these large, branched systems. Is there any indication of whether there are interactions between pillararenes on adjacent branches? The authors should look at NOESY NMR of these dendrimer species to evaluate inter-branch interactions.

Reply: According to the reviewer’s suggestion, the additional 2-D ROESY NMR analysis of **G₁-G₃** was performed to evaluate the possible interactions between pillararenes on adjacent branches. As shown in Fig. R25-R30, due to the overlap of specific peaks attributed to the pillararene macrocycles, no obvious correlations were distinguished as inter-branch interactions from intra-branch interactions. Actually, although the third-generation rotaxane-branched dendrimer in Figure S79 were drawn in a two-dimensional form, high-generation rotaxane-branched dendrimer usually features three dimensional characters.

Fig. R25 2-D ROESY spectrum of rotaxane-branched dendrimer **G₁**.

Fig. R26 2-D ROESY spectrum of rotaxane-branched dendrimer **G₁** with the addition of TBAA (5.0 eq. for each urea unit).

Fig. R27 2-D ROESY spectrum of rotaxane-branched dendrimer **G₂**.

Fig. R28 2-D ROESY spectrum of rotaxane-branched dendrimer G_2 with the addition of TBAA (5.0 eq. for each urea unit).

Fig. R29 2-D ROESY spectrum of rotaxane-branched dendrimer G_3 .

Fig. R30 2D-ROESY spectrum of rotaxane-branched dendrimer G_3 with the addition of TBAA (5.0 eq. for each urea unit).

Overall, the paper is interesting but needs additional experiments and evaluation prior to any possible publication.

Reply: Again, we greatly appreciate the reviewer's positive comments to the chemistry presented in the manuscript as well as these thorough suggestions, which are definitely helpful to improve the quality of this manuscript. Following the reviewer's advices, we have accordingly revised the manuscript to address all the issues that listed by the reviewer.

Reviewer #3 (Remarks to the Author):

"Dual Stimuli-responsive Rotaxane-branched Dendrimers with Reversible Dimension Modulation"

The manuscript by Yang and co-workers build a series poly-[2]-rotaxane containing organometallic dendrimers and examine the increase and decrease of the size of the dendrimers when treated with dimethylsulfoxide (DMSO) molecules or acetate anions. Initially, they report the synthesis and

switching behaviour of the “simple” organometallic [2]rotaxane. The system is well characterised using NMR, MS and calculations. It would be nice if the authors provided the energies of the calculated structures but from the NMR data it is clear that the switching occurs.

Reply: We greatly appreciate the reviewer’s positive comments to the chemistry presented in this manuscript. According to the reviewer’s suggestion, the energies of the calculated structure of the organometallic [2]rotaxane **2** were obtained from the DFT calculation (Fig. R31). The discussion about the calculated structure of [2]rotaxane **2** has been revised in main text of the revised manuscript. Moreover, the calculated structures with the energies have been presented in Supporting Information.

Fig. R31 Theoretically calculated structures with energies of (a) [2]rotaxane **2**; (b) solvent-induced and (c) anion-induced switching motion of DEP5 ring in [2]rotaxane **2**.

The authors then use the organometallic [2]rotaxane as a building block to generate the branched dendrimers, using a divergent approach allowed the successful synthesis of a new family of rotaxane-branched dendrimers up to a third generation system which contained twenty-one switchable rotaxane moieties.

Again the dendrimers were well characterised using NMR, MS, GPC and AFM, TEM was less convincing, but DOSY NMR also confirmed the formation and size of the systems. The authors then repeated the switching experiments and showed that the addition and removal of DMSO or acetate anions resulted in switching of the macrocycle along the rotaxane thread (as observed in the simpler material). They also found that the size of the dendrimers changes when treated with the external

stimuli, and suggest that the movement of the macrocycle is amplified in the dendrimers generating the expansion and contraction of the dendrimers. This could be correct but the authors have not done the controls that would confirm their hypothesis. The expansion and contraction (swelling/deswelling) of polymer, gels and dendrimers is known to occur when the solvent or anions are changed thus the observed size change may just be due to the swelling/deswelling of the dendrimers and not connected to the movement of the macrocycle. To prove/disprove this the authors need to make the series of analogues dendrimers that do not feature the macrocycle and examine the switching and the size change. This must be done before publication as the current work does not provide enough evidence that the change in size of the system is related to the switching of the rotaxanes.

Reply: We agree with the reviewer's opinion about the control experiments. Thus according to the reviewer's advice, the model first-generation dendrimers either without urea moiety (**G1-a**) or without macrocycles moiety (**G1-b**) were synthesized by employing the same controllable divergent approach from the corresponding model complexes **2-a** or **2-b**, respectively (Scheme R4 and R5). All the model complexes (**2-a**, **2-b**, **G1-a**, and **G1-b**) were well-characterized by multinuclear NMR (^1H , ^{13}C , and ^{31}P) spectroscopy and MS analysis. Subsequently, the investigations on stimuli-responsive behavior of these model dendrimers were carried out to confirm that the size modulation of the integrated rotaxane-branched dendrimers was attributed to the rotaxane switching on each branch. In the control experiments, both DMSO molecule and acetate anion were selected as external stimuli to study the stimuli-responsive behavior.

In the case of model complex **2-a** without urea moiety, upon the addition of DMSO- d_6 (10 μL) or TBAA (5.0 eq.) into the solution of **2-a** in THF- d_8 (9.0 mM, 400 μL), the resultant spectra showed almost no change compared with the original ones as indicated in ^1H NMR spectra (Fig. R32). This observation indicated that the urea moiety in [2]rotaxane **2** did act as a binding site. Similar with the model model complex **2-a**, in the case of the model dendrimer **G1-a** without urea moiety, due to the absence of binding site, no obvious change was found in the ^1H NMR spectra upon the addition of either DMSO molecule or acetate anion as stimulus (Fig. R33). Moreover, for model complex **2-b** without pillar[5]arene as the wheel, upon adding either DMSO- d_6 (10 μL) or TBAA (5.0 eq.) into the solution of **2-b** in THF- d_8 (9 mM, 400 μL), the protons of urea moiety (H_3 and H_4) displayed remarkable downfield shifts as shown in ^1H NMR spectra (Fig. R34), indicating the existence of hydrogen bonding interactions between the urea moiety with either DMSO molecule or acetate anions, respectively.

Similar with the model model complex **2-b**, in the case of model rotaxane dendrimer without macrocycles moiety (**G1-b**), upon the addition of either DMSO molecule or acetate anion as stimulus, the existence of hydrogen bonding interactions between the urea moiety with either DMSO molecule

or acetate anion was confirmed as evidenced by the remarkable chemical shifts of H₃ and H₄ in the ¹H NMR spectrum (Fig. R35). Since the acetate anion is a better stimulus than DMSO molecule as evidence by the larger downfield shifts in the ¹H NMR spectrum, acetate anion was selected as an external stimulus to study the size modulation property of model dendrimers by using 2-D DOSY technique. It was found that, for both model dendrimers, almost no change of the diffusion coefficient (*D*) value before and after the addition of anion was found (for **G_{1-a}**, $D = (13.11 \pm 0.07) \times 10^{-10} \text{ m}^2/\text{s}$, for the mixture of **G_{1-a}** and TBAA, $D = (13.26 \pm 0.08) \times 10^{-10} \text{ m}^2/\text{s}$; for **G_{1-b}**, $D = (9.83 \pm 0.06) \times 10^{-10} \text{ m}^2/\text{s}$, for the mixture of **G_{1-b}** and TBAA, $D = (9.72 \pm 0.05) \times 10^{-10} \text{ m}^2/\text{s}$) (Fig. R36-41). In order to provide further information on stimuli-responsive behavior of model dendrimers in solution phase, dynamic light scattering (DLS) measurement was performed, which revealed almost no size change of both model dendrimers maintained before and after adding 5.0 eq. of TBAA (For **G_{1-a}**, before: 1.35 nm, after: 1.39 nm; for **G_{1-b}**, before: 1.55 nm, after: 1.53 nm) as shown in Fig. R42. These observations clearly demonstrated that the size of the model dendrimers didn't change with the addition of anion, which might exclude the anion effect that caused the swelling/de-swelling of the rotaxane-branched dendrimers.

The discussion about the additional model experiments was added in main text of the revised version. Moreover, the additional model experiment results have been provided in Supplementary Information.

Scheme R4. The synthesis routes of (a) [2]rotaxane building block **2-a** without urea moiety as the stimuli-responsive site and (b) model first-generation rotaxane dendrimer **G_{1-a}** from corresponding building block **2-a**.

Scheme R5. The synthesis routes of (a) platinum-acetylide building block **2-b** without pillar[5]arene as the wheel and (b) model first-generation platinum-acetylide dendrimer **G_{1-b}** from corresponding building block **2-b**.

Fig. R32 ¹H NMR spectra (THF-*d*₈, 298 K, 500 MHz) of model complex **2-a** with the addition of DMSO-*d*₆ (10 μL) (*bottom*); **2-a** (*middle*); **2-a** with the addition of TBAA (5.0 eq.) (*top*).

Fig. R33 ^1H NMR spectra (THF- d_8 , 298 K, 500 MHz) of **G₁-a** with the addition of DMSO- d_6 (10 μL) (bottom); model dendrimer **G₁-a** (middle); **G₁-a** with the addition of 15.0 eq. of TBAA (top).

Fig. R34 ^1H NMR spectra (THF- d_8 , 298 K, 500 MHz) of **2-b** with the addition of DMSO- d_6 (10 μL) (bottom); model complex **2-b** (middle); **2-b** with the addition of 5.0 eq. of TBAA (top).

Fig. R35 ^1H NMR spectra ($\text{THF-}d_8$, 298 K, 500 MHz) of $\text{G}_1\text{-b}$ with the addition of $\text{DMSO-}d_6$ (10 μL) (*bottom*); model dendrimer $\text{G}_1\text{-b}$ (*middle*); $\text{G}_1\text{-b}$ with the addition of 15.0 eq. of TBAA (*top*).

Fig. R36 2-D DOSY spectrum ($\text{THF-}d_8$, 298 K, 500 MHz) of model rotaxane dendrimer $\text{G}_1\text{-a}$.

Fig. R37 2-D DOSY spectrum (THF-*d*₈, 298 K, 500 MHz) of model rotaxane dendrimer **G1-a** with the addition of 15.0 eq. of TBAA.

Fig. R38 Stacked DOSY spectra (THF-*d*₈, 298 K, 500 MHz) of model rotaxane dendrimer **G1-a** (red) and **G1-a** with the addition of 15.0 eq. of TBAA (blue).

Fig. R39 2-D DOSY spectrum (THF-*d*₈, 298 K, 500 MHz) of model platinum-acetylide dendrimer **G1-b**.

Fig. R40 2-D DOSY spectrum (THF-*d*₈, 298 K, 500 MHz) of model platinum-acetylide dendrimer **G1-b** with the addition of 15.0 eq. of TBAA.

Fig. R41 Stacked 2-D DOSY spectra (THF-*d*₈, 298 K, 500 MHz) of model platinum-acetylide dendrimer **G1-b** (red) and **G1-b** with the addition of 15.0 eq. of TBAA (blue).

Fig. R42 DLS spectra of (a) model rotaxane dendrimer **G1-a** and the mixture of **G1-a** with the addition of 15.0 eq. of TBAA; (b) model dendrimer **G1-b** and the mixture of **G1-b** with the addition of 15.0 eq. of TBAA.

Additionally, some of the English need rewording and the some of the pictures also need to fixed! The resolution of Scheme 1 and Fig.1 is poor and need to be fixed.

The X axis on the NMR spectra is impossible to read and for some reason seem to say (f1) ppm rather than the correction delta (ppm).

The resolution of 5(a) and 6(a-f) needs to be improve the blue lines in 6 should be made thicker.

The histogram is figure 5 is not clear what are all the different colors referring to?

Reply: According to the reviewer's suggestion, the English writing has been well polished throughout the whole manuscript and all above-mentioned errors have been revised. In addition, during the doc-pdf conversion process of the first round submission, most of figures became fuzzy. In the revised manuscript, all figures were re-edited and submitted in high-resolution form.

The authors should also state somewhere that addition of Na⁺ in THF causes the precipitation of NaOAc (s) in order to remove the OAc anions from solution as this was not immediately clear in the manuscript

Reply: According to the reviewer's suggestion, the detailed description of the anion removal by using sodium salt was added in the main text as follows.

"In order to completely remove the acetate anion as NaOAc precipitate, 7.0 equiv. of NaPF₆ was subsequently added into the mixture of [2]rotaxane 2 and TBAA".

This is certainly nice interesting work and it could be great paper once the control experiments are done.

Therefore, I recommend that the manuscript be rejected then resubmitted after the correct controls reactions are carried out and the other more minor change are attended to.

Reply: We greatly appreciate the reviewer's positive comments as well as the insightful advices to the chemistry presented in this manuscript. According to the reviewer's suggestions, we have already revised our manuscript and addressed all the issues listed by the reviewer.

Again we deeply appreciate all reviewers' thoughtful suggestions that obviously improved the quality of the manuscript. Considering all reviewers' positive comments on the novelty and significance of the chemistry presented in this manuscript, with these changes and responses we hope the revised manuscript is now acceptable for publication in *Nature Communications*.

With many thanks and best regard.

Hai-Bo Yang

Reviewers' comments:

Reviewer #1 (Remarks to the Author):

The authors have addressed the issue with the switching of the system by providing the control dumbbell experiment. They have also given some error data as requested.

However there are still some concerns - using DOESY to show purity is unheard of. 1.03 PDI is close but not pure, and the elemental analysis numbers are off.

More importantly it is clear from the DOESY and DLS experiments that after one switching cycle the size does not go back to original state and there is no discussion about this in the letter. More importantly, AFM and DLS are showing sizes in the range of 2-3 of a compound having multiple pillarenes on it - each one of these rings is 1 nm in size. How can 20 of them condense to give a dynamic radius of 2-3 is not clear (the reference to a polymeric system which is not as rigid as the rings is not a reasonable one)

Reviewer #2 (Remarks to the Author):

The authors have provided a robust response to the comments of the various reviewers and in the opinion of this reviewer have addressed the many comments that have been raised to an appropriate level. I am happy with the changes that have been made and now recommend acceptance of the article.

One small point I would like to see revised is that in response to reviewer 1 the authors report the CHN for G1. I would like to know how the authors justify the elemental analysis for [G1 + 3CH₂Cl₂]. What is the evidence for 3 CH₂Cl₂? It isn't observed in the ¹H or ¹³C NMR (Fig S25/S27). Personally I think CHN for such large compounds is not helpful as there can be many issues with using this technique for such large systems. I would prefer that the authors do not report the data rather than simply adding in CH₂Cl₂ molecules, if this is what has happened. I recommend that the CHN data is removed but I would imagine that other reviewers may disagree. I leave this to the judgement of the editor.

Reviewer #3 (Remarks to the Author):

Review for NCOMMS-17-31552A

Dual Stimuli-responsive Rotaxane-branched Dendrimers with Reversible Dimension Modulation

The authors have described their efforts to construct a series of dual stimuli-responsive rotaxane-branched dendrimers. Using a switchable organometallic [2]rotaxane precursor, three rotaxane-branched dendrimers up to the third generation system with twenty-one switchable rotaxane moieties located on each branch were generated. More importantly, upon the addition and removal of dimethylsulfoxide (DMSO) molecule or acetate anion as the external stimulus, the motion of the macrocycle on the switchable rotaxane units results in a size change for the rotaxane-branched dendrimers, thus leading to the dimension modulation of the materials.

The authors have examined the advice provided by the reviewers and significantly improved the manuscript. Their characterization of the rotaxane and the dendrimers was initially good but as requested by all the referees additional experiments (GPC and DOSY) were carried out to give more information of the purity and size of the dendrimers.

The manuscript is accompanied but a comprehensive supporting information document that provides all the data acquired by the authors. As requested by referee 1 the authors have added error bars/ uncertainties into the manuscript and the SI to help the readers.

Most importantly, the authors have made some new materials to use as controls and showed that the size change of the branched dendrimers is not simply caused by a solvent or anion swelling effect the change does required the present of the macrocycle and the presumably the stimuli induced motion.

It is still not 100% clear why the motion of the macrocycle causes the size change but the authors suggest that the ...“location of DEP5 rings on the axle...is influencing the rigidity of all branches” and this seem plausible.

There are two small things that need addressing before publication. The authors say on page six that the ^1H NMR titration data for 2 was fitted to a 2:3 binding model (why? What data do they have to suggest this HG ratio? ESI-MS?? Mole ratio method?? Job plot is suggested in the paper and the SI but the use of this kind of plot has been shown to be untrustworthy (see Chem. Commun., 2016,52, 12792-12805 and references within)

As far as I can see the rotaxane 2 only has one binding site so I would expected a 1:1 HG complex for this compound. Why do the authors think a 2:3 model is appropriate?

Also how did the authors get the three (K_1 , K_2 and K_3) binding constants from the data? Most ^1H titration programs only fit 1:1 and 1:2 binding models because higher H-G ratios cannot be obtained due to the large number of variables that need to be fitted (see

<http://supramolecular.org/>, or

<https://community.dur.ac.uk/j.m.sanderson/science/downloads.html>, Chem. Soc. Rev., 2011, 40, 1305–1323. So what equations were used to get the fit and what program was used?

Response to Referee 1:

The authors have addressed the issue with the switching of the system by providing the control dumbbell experiment. They have also given some error data as requested.

However there are still some concerns - using DOESY to show purity is unheard of.

Reply: We fully understand the reviewer's concern on DOSY measurement. DOSY is a well-established two-dimensional ^1H NMR technique for characterization of complex compounds. Recently, DOSY has evolved to be a highly efficient method to evaluate the purity of the higher-order complicated systems such as dendrimers, polymers, and metallosupramolecular architectures, which has been widely employed by many groups. The followings are the related reports.

"The objective is thus double: to measure the diffusion coefficients of the molecules in solution and to obtain a DOSY spectrum that reflects the purity of the assembly." (D. Astruc *et al. Chem. Eur. J.*, **2008**, *14*, 5577-5587; *Chem. Eur. J.*, **2014**, *20*, 11176 -11186.)

"The solution of catenated species in a mixture of $[\text{D}_6]\text{DMSO}/\text{D}_2\text{O}$ (1:1) was subjected to a DOSY (diffusion ordered NMR spectroscopy) study to confirm the purity of the component." (M. Fujita *et al. Angew. Chem. Int. Ed.*, **2005**, *44*, 4896-4899.)

"2D DOSY-NMR spectroscopy has frequently been utilized to determine the purity of the complexes due to its effective characterization for molecules having relatively large molecular weights and for monitoring self-assembly processes by correlating chemical resonances to diffusion coefficients in solutions." (P. Wang *et al. RSC Adv.*, **2016**, *6*, 5631-5635.)

"Furthermore, DOSY NMR experiments are being more and more implemented to evidence the formation of block copolymers and to assess their purity." (S. M. Guillaume *et al. Polym. Chem.*, **2016**, *7*, 4603-4608.)

*"In order to investigate the behaviour and purity of these species in solution, DOSY NMR studies in CD_2Cl_2 at room temperature were performed on helicates 4b-6b (ESI $^+$)." (S. A. Baudron, M. W. Hosseini *et al. Chem. Commun.*, **2015**, *51*, 5906-5909.)*

"Diffusion-ordered spectroscopy (DOSY) (Figure S4) showed that the signals for the aromatic and Cp units displayed similar diffusion constants, which further confirmed the purity of 5a in methanol."* (G.-X. Jin *et al. Chem*, **2017**, *3*, 110-121.)

*“A single diffusion coefficient in the DOSY NMR ($D = 4.5 \times 10^{-10} \text{ m}^2 \text{ s}^{-1}$) as well as a single set of signals in the ^1H -NMR provided evidence for its purity.” (M. Schmittel *et al. Chem. Commun.*, **2016**, 52, 8749-8752.)*

*“In addition, DOSY NMR was used to confirm the purity of the products (King *et al.*, 2010).” (S. R. Labafzadeh *et al. Carbohydr. Polym.*, **2015**, 116, 60-66.)*

*“To verify the topology and purity of the comb polymer, diffusion-ordered NMR spectroscopy (DOSY) was utilized (Fig. 4c).” (B. S. Sumerlin *et al. Nat. Chem.*, **2017**, 9, 817-823.)*

1.03 PDI is close but not pure, and the elemental analysis numbers are off.

Reply: For all rotaxane-branched dendrimers **G₁-G₃**, the careful workups by employing column chromatography and preparative gel permeation chromatography (GPC) were performed to get the targeted products in high purity. Notably, during the purification process by using preparative GPC, all products displayed a single dominant peak (as shown in *SI Appendix*, Fig. S24), thus indicating the existence of one fraction. Furthermore, in order to enhance the purity, the repeatable dissolution-precipitation processes were conducted. The purity of the targeted products was firstly checked by one-dimensional (1-D) ^1H and ^{31}P NMR spectroscopy, in which one set of clearly-assigned signal was observed in each spectrum. Moreover, for all three rotaxane-branched dendrimers, only one set of signal was observed in 2-D DOSY spectra, thus indicating the existence of the sole specie. From a synthetic organic chemist point of view, the combination of aforementioned results strongly supports the high purity of the targeted rotaxane-branched dendrimers.

Considering the reviewer’s concern on PDI, in the revised manuscript, PDI is considered as an indicator for the monodispersity (*‘a key feature of dendrimers’*) rather than the purity, which now might avoid the possible misunderstanding. Thus the related description was modified as *“Gel permeation chromatography (GPC) experiments were then carried out to confirm the formation as well as the monodispersity of rotaxane-branched dendrimers. In the GPC spectra (Fig. S47-49), all rotaxane-branched dendrimers exhibited a single peak and narrow distributions for the number-averaged molecular weight (M_n) and the polydispersity index (PDI) (for **G₁**, PDI = 1.03; for **G₂**, PDI = 1.04; for **G₃**, PDI = 1.15), indicating the existence of monodisperse rotaxane-branched dendrimers **G₁-G₃**. ”*

In the case of element analysis, given the existence of huge molecular weights and numbers of rotaxane units, it is really difficult to obtain the satisfied element analysis data even after continuous attempts. Notably, as pointed out by the second reviewer, CHN analysis for our rotaxane-branched

dendrimers with large skeleton is not helpful because many issues would exist when using such technique. So the element analysis data was deleted in the revised version.

More importantly it is clear from the DOESY and DLS experiments that after one switching cycle the size do not go back to original state and there is no discussion about this in the letter.

Reply: It is true that, in the DOSY and DLS analysis, the size cannot go back to the original state due to the *in situ* formation of NaOAc precipitate that might slightly influence the micro-environment. However, as indicated by both ¹H NMR titration experiments and recycling experiments, after one switching cycle, all rotaxane-branched dendrimers could fully go back to their original state, which demonstrated the reversible stimuli-responsive motion of rotaxane-branched dendrimers.

According to the reviewer's suggestion, in the revised version, the related discussion on DOSY and DLS experiments has been revised/added as follows.

For DOSY analysis: *“Notably, although the diffusion coefficients could not return to the original values possibly due to the existence of the in situ formed NaOAc precipitate, the trend of the size switching was reasonable.”* (Page 8, left column, highlight in yellow).

For DLS analysis: *“Similar with 2-D DOSY analysis, the sizes of the rotaxane-branched dendrimers could not fully go back to the original state, which might be caused by the existence of the in situ formed NaOAc precipitate.”* (Page 8, right column, highlight in yellow).

More importantly, AFM and DLS are showing sizes in the range of 2-3 of a compound having multiple pillarenes on it - each one of these rings is 1 nm in size. How can 20 of them condense to give a dynamic radius of 2-3 is not clear (the reference to a polymeric system which is not as rigid as the rings is not a reasonable one)

Reply: We fully understand the reviewer's concern on the relatively small sizes of such huge rotaxane-branched dendrimers in the AFM and DLS analysis. We have repeated the experiments for several times and the similar results were obtained.

In order to obtain the reasonable explanation for the obtained results, we have already studied the related literatures very carefully. Actually we are not the first one who gets the relatively small sizes of such huge dendrimers by using AFM and DLS techniques. For example, in the case of classical dendrimer PAMAM (*J. Am. Chem. Soc.*, **1998**, *120*, 5323.), as the reviewer pointed out, the skeleton is not as rigid as our rotaxane-branched dendrimers. However, there are thousands of repeating units

in the eighth-generation PAMAM and the height range was found from 3.5 to 4.0 nm. It should be noted that, in AFM measurement, only the height information of dendrimers on the surface rather than the ideal-sphere size can be obtained. Thus it is reasonable that the value is much smaller than the ideal-sphere diameter due to the surface-induced de-conformation and/or structural collapse by the solvent loss. In the case of rotaxane-branched dendrimers in this study, there are only twenty-one rotaxane branches in the dendrimer skeleton of G_3 . Notably, these branches are not connected in a head-to-tail fashion, which were distributed into three different generations of the dendrimer skeleton in a monodispersed way. In addition, these rotaxane moieties are neither fully extended nor standing right on the surface. Thus the height 3.21 ± 0.34 nm is reasonable. In the case of DLS analysis, in order to confirm the reproducibility of the size values, we re-performed the DLS measurements for several times. As shown in Fig. R1, after several attempts, the similar size information was obtained. Considering the flexible feature of alkyl chain in the axle, we assumed that rotaxane-branched dendrimers were neither fully extended nor an ideal-sphere structure, which might lead to the relatively small size in the DLS analysis.

Fig. R1 The repeated DLS spectra of rotaxane-branched dendrimer G_3 . The size of G_3 is (a) 4.51 nm, (b) 4.35 nm, (c) 4.48 nm, (d) 4.88 nm.

Response to Referee 2:

The authors have provided a robust response to the comments of the various reviewers and in the opinion of this reviewer have addressed the many comments that have been raised to an appropriate level. I am happy with the changes that have been made and now recommend acceptance of the article.

One small point I would like to see revised is that in response to reviewer 1 the authors report the CHN for **G**₁. I would like to know how the authors justify the elemental analysis for [**G**₁ + 3CH₂Cl₂]. What is the evidence for 3 CH₂Cl₂? It isn't observed in the ¹H or ¹³C NMR (Fig S25/S27). Personally I think CHN for such large compounds is not helpful as there can be many issues with using this technique for such large systems. I would prefer that the authors do not report the data rather than simply adding in CH₂Cl₂ molecules, if this is what has happened. I recommend that the CHN data is removed but I would imagine that other reviewers may disagree. I leave this to the judgement of the editor.

Reply: We fully agree with the reviewer on his/her comments on element analysis. Indeed, considering the huge molecular weights and numbers of rotaxane units of rotaxane-branched dendrimers, it is really difficult to obtain the satisfied element analysis data even after continuous attempts. In addition, for the large molecules with giant skeleton, it is common to trap solvent molecules within their scaffolds. In the case of **G**₁, three CH₂Cl₂ molecules were assumed to be encapsulated in each rotaxane-branched dendrimer because CH₂Cl₂ was used in the workup process.

Actually, the purity of the targeted products was firstly checked by one-dimensional (1-D) ¹H and ³¹P NMR spectroscopy, in which one set of clearly-assigned signal that attributed to the sole component was observed in each spectrum. Moreover, for all three rotaxane-branched dendrimers, only one set of signal was observed in 2-D DOSY spectra, thus indicating the existence of the sole specie. From a synthetic organic chemist point of view, the combination of aforementioned results strongly supports the high purity of the targeted rotaxane-branched dendrimers.

As pointed out by the reviewer, CHN analysis for our rotaxane-branched dendrimers with large skeleton is not helpful because many issues would exist when using such technique. So the element analysis data was deleted in the revised version.

Response to Referee 3:

Review for NCOMMS-17-31552A Dual Stimuli-responsive Rotaxane-branched Dendrimers with Reversible Dimension Modulation. The authors have described their efforts to construct a series of dual stimuli-responsive rotaxane-branched dendrimers. Using a switchable organometallic [2]rotaxane precursor, three rotaxane-branched dendrimers up to the third generation system with twenty-one switchable rotaxane moieties located on each branch were generated. More importantly, upon the addition and removal of dimethylsulfoxide (DMSO) molecule or acetate anion as the external stimulus, the motion of the macrocycle on the switchable rotaxane units results in a size change for the rotaxane-branched dendrimers, thus leading to the dimension modulation of the materials.

The authors have examined the advice provided by the reviewers and significantly improved the manuscript. Their characterization of the rotaxane and the dendrimers was initially good but as requested by all the referees additional experiments (GPC and DOSY) were carried out to give more information of the purity and size of the dendrimers.

The manuscript is accompanied by a comprehensive supporting information document that provides all the data acquired by the authors. As requested by referee 1 the authors have added error bars/uncertainties into the manuscript and the SI to help the readers.

Most importantly, the authors have made some new materials to use as controls and showed that the size change of the branched dendrimers is not simply caused by a solvent or anion swelling effect; the change does require the presence of the macrocycle and the presumably the stimuli induced motion.

It is still not 100% clear why the motion of the macrocycle causes the size change but the authors suggest that the ...“location of DEP5 rings on the axle...is influencing the rigidity of all branches” and this seems plausible.

There are two small things that need addressing before publication. The authors say on page six that the ¹H NMR titration data for **2** was fitted to a 2:3 binding model (why? What data do they have to suggest this HG ratio? ESI-MS?? Mole ratio method?? Job plot is suggested in the paper and the SI but the use of this kind of plot has been shown to be untrustworthy (see Chem. Commun., 2016,52, 12792-12805 and references within) As far as I can see the rotaxane **2** only has one binding site so I

would expect a 1:1 HG complex for this compound. Why do the authors think a 2:3 model is appropriate?

Reply: We fully agree with the reviewer's comment on the expected 1:1 binding model between rotaxane **2** and acetate anion. The Job plot method has been one of the most popular and generally-accepted methods for determining the stoichiometry in host-guest chemistry. However, as pointed out by the reviewer, recent investigations indicated that the Job plot experiment is influenced by various factors such as concentration, self-aggregation, and/or ionic strength *etc.* Thus the stoichiometry for the same host-guest system under the varied titration conditions might be different. According to the previous titration experiments, a 2:3 binding model could be regarded as the combination of 1:1 and 1:2 binding model (C.-L. Wang, L. Zhou, L. Zhang, J.-F. Xiang, B. M. Rambo, J. L. Sessler and H.-Y. Gong, *Chem. Commun.*, **2017**, 53, 3669; M. Anioła, Z. D. Szafran, A. Katrusiak, A. Komasa and M. Szafran, *Chem. Phys.*, **2016**, 477, 88.).

According to the reviewer's suggestion, we repeated the titration at a relatively diluted concentration (from 2.0 mM in previous titration to 0.5 mM in current titration), which might reduce the concentration effect. To our delight, according to the new titration data, a 1:1 binding model was obtained with a binding constant $\log K = 3.57 \pm 0.2$ (Fig. R2 and R3). On the basis of the structural feature of rotaxane **2** as well as the common binding model between urea moiety and acetate anion, a 1:1 binding model is more reasonable. Thus in the revised manuscript, the previous titration data was replaced by the updated one as “*With the aim to obtain the further insight into the anion-induced switching of [2]rotaxane 2, the acetate binding affinity of [2]rotaxane 2 was determined by ¹H NMR titrations with acetate anion (TBAA) in THF-d₈. The data was fitted to a 1 : 1 binding model (2: acetate anion) as confirmed by Job plot analysis, and the anion binding constant was calculated to be $\log K = 3.57 \pm 0.2$ (Fig. S78).*”

Fig. R2 The expanded region of the ^1H NMR spectra (THF- d_8 , 298 K, 500 MHz) of [2]rotaxane **2** (Host) with the addition of acetate anion (Guest).

Fig. R3 (a) Job plot for [2]rotaxane **2**-acetate anion complex in THF- d_8 ($[2] + [\text{anion}] = 1 \text{ mM}$); (b) The ^1H NMR titration isotherm of [2]rotaxane **2** with the addition of acetate anion (TBAA) recorded at 500 MHz in THF- d_8 at 298 K. (●, ◀, ▶, ●, ★ indicate the change in chemical shift of the proton signals corresponding to H₁, H₁₃, H₁₆, H₁₄, H₁₅, respectively, on rotaxane **2** in Fig. R2).

Also how did the authors get the three (K_1 , K_2 and K_3) binding constants from the data? Most ^1H titration programs only fit 1:1 and 1:2 binding models because higher H-G ratios cannot be obtained due to the large number of variables that need to be fitted (see <http://supramolecular.org/>, or <https://community.dur.ac.uk/j.m.sanderson/science/downloads.html>, Chem. Soc. Rev., 2011, 40, 1305–1323). So what equations were used to get the fit and what program was used?

Reply: In the case of rotaxane-branched dendrimer \mathbf{G}_1 , the titration data was fitted to a 1 : 3 binding model (\mathbf{G}_1 : acetate anion). On the basis of our titration data, by using the Hyperquad 2003 program (Hyperquad 2003: P. Gans, A. Sabatini, A. Vacca, *Talanta*, **1996**, 43, 1739.), the equilibrium constants (K_1 , K_2 , K_3) of \mathbf{G}_1 (H) and acetate anion (G) were obtained, as listed below:

According to the equilibrium constants K , the three binding constants K' were calculated, the details are shown below:

$$K_1' = K_1 = [\text{HG}] / ([\text{H}] [\text{G}])$$

$$\text{Thus, } \log K_1' = \log K_1 = 4.19$$

$$K_2' = [\text{HG}_2] / ([\text{HG}] [\text{G}]) = (K_2 [\text{H}] [\text{G}]^2) / (K_1 [\text{H}] [\text{G}] [\text{G}]) = K_2 / K_1$$

$$\text{Thus, } \log K_2' = \log K_2 - \log K_1 = 7.62 - 4.19 = 3.43$$

$$K_3' = [\text{HG}_3] / ([\text{HG}_2] [\text{G}]) = (K_3 [\text{H}] [\text{G}]^3) / (K_2 [\text{H}] [\text{G}]^3) = K_3 / K_2$$

$$\text{Thus, } \log K_3' = \log K_3 - \log K_2 = 10.83 - 7.62 = 3.21$$

The calculation details have been added in Supplementary Information.

REVIEWERS' COMMENTS:

Reviewer #1 (Remarks to the Author):

Again DOSY cannot be used to show purity of composition but the fact that there are self-assembled structures having the same size. All the cited papers are doing that.

The AFM measurements are not helpful if they cannot give proper characterization of the compound!

Reviewer #2 (Remarks to the Author):

Overall, I find that the authors have addressed the points raised by the reviewers. I was slightly concerned by their use of DOSY to suggest purity. I think the authors do acknowledge that the DOSY spectrum evaluates the speciation rather than purity, i.e. there is a single species present in solution. This is different from purity as there is a possibility that the dendrimer is not pure but retains the same overall size.

I would suggest that the authors modify the sentence "Moreover, 2-D diffusion-ordered spectroscopy (DOSY)47-50 was also exploited to evaluate the purity and size change..." to say something like "Moreover, 2-D diffusion-ordered spectroscopy (DOSY)47-50 was also exploited to evaluate the monodispersity and size change...".

Otherwise I find the appear to be acceptable for publication.

Reviewer #3 (Remarks to the Author):

Review for manuscript NCOMMS-17-31552B

The authors have suitably addressed my queries around the binding constants and binding models, they also seem to have answered the queries of the other referees. The paper seem suitable for publication.

Reviewer #1 (Remarks to the Author):

Again DOESY cannot be used to show purity of composition but the fact that there are self-assembled structures having the same size. All the cited papers are doing that. The AFM measurements are not helpful if they cannot give proper characterization of the compound!

Reply: We fully understand the reviewer's concern on DOSY measurement. In order to avoid the possible misunderstanding, as suggested by Referee 2, DOSY analysis is considered as a technique to evaluate the monodispersity of rotaxane-branched dendrimer rather than the purity in the revised version. Thus the related description was modified as "*Moreover, 2-D diffusion-ordered spectroscopy (DOSY)⁴⁷⁻⁵⁰ was also exploited to evaluate the monodispersity and size change of the resultant rotaxane-branched dendrimers G₁-G₃.*".

In addition, we agree with the reviewer that AFM measurements are not helpful if they cannot give proper characterization of the compound. In this study, AFM measurement was employed to provide morphology information of the resultant rotaxane-branched dendrimers together with the TEM analysis. Actually, AFM has been recently used to study the morphology of complicated supramolecular assemblies as shown in many literatures (B.-P. Jiang, D.-S. Guo, Y.-C. Liu, K.-P. Wang and Y. Liu, *ACS Nano*, **2014**, 8, 1609; A. Takai, T. Kajitani, T. Fukushima, K. Kishikawa, T. Yasuda and M. Takeuchi, *J. Am. Chem. Soc.*, **2016**, 138, 11245; N. Avakyan, A. A. Greschner, F. Aldaye, C. J. Serpell, V. Toader, A. Petitjean and H. F. Sleiman, *Nat. Chem.*, **2016**, 8, 368; T. Fukui, S. Kawai, S. Fujinuma, Y. Matsushita, T. Yasuda, T. Sakurai, S. Seki, M. Takeuchi and K. Sugiyasu, *Nat. Chem.*, **2017**, 9, 493; G.-Q. Yin, H. Wang, X.-Q. Wang, B. Song, L.-J. Chen, L. Wang, X.-Q. Hao, H.-B. Yang, X. Li, *Nat. Commun.*, **2018**, 9, 567.)

We again greatly appreciate the thorough and constructive suggestion of the Referee 1, which obviously improved the quality of this manuscript.

Reviewer #2 (Remarks to the Author):

Overall, I find that the authors have addressed the points raised by the reviewers. I was slightly concerned by their use of DOSY to suggest purity. I think the authors do acknowledge that the DOSY spectrum evaluates the speciation rather than purity, i.e. there is a single species present in solution. This is different from purity as there is a possibility that the dendrimer is not pure but retains the same overall size.

I would suggest that the authors modify the sentence “Moreover, 2-D diffusion-ordered spectroscopy (DOSY)⁴⁷⁻⁵⁰ was also exploited to evaluate the purity and size change...” to say something like “Moreover, 2-D diffusion-ordered spectroscopy (DOSY)⁴⁷⁻⁵⁰ was also exploited to evaluate the monodispersity and size change...”.

Otherwise I find the appear to be acceptable for publication.

Reply: We fully agree with the reviewer on his/her comments on DOSY analysis. In order to avoid the possible misunderstanding, in the revised version, DOSY analysis is considered as a technique to evaluate the monodispersity of rotaxane-branched dendrimer rather than the purity. Thus the related description was modified as “*Moreover, 2-D diffusion-ordered spectroscopy (DOSY)⁴⁷⁻⁵⁰ was also exploited to evaluate the monodispersity and size change of the resultant rotaxane-branched dendrimers G₁–G₃.*”.

We greatly appreciate the thorough and constructive suggestion of the Referee 2, which obviously improved the quality of this manuscript.

Reviewer #3 (Remarks to the Author):

Review for manuscript NCOMMS-17-31552B

The authors have suitably addressed my queries around the binding constants and binding models, they also seem to have answered the queries of the other referees. The paper seems suitable for publication.

Reply: Again we greatly appreciate the thorough and constructive suggestion of the Referee 3, which obviously improved the quality of this manuscript.